# Localized transmission of an aquatic pathogen drives hidden epidemics and population collapse in a terrestrial host

**Andrés Valenzuela-Sánchez** [1,2,3] ✉, **Soledad Delgado-Oyarzún**[2], **Claudio Azat** [4], **Benedikt R. Schmidt** [5,6], **Hugo Sentenac** [7], **Natashja Haddow**[2,4,8], **Bastián Santana**[2], **Jaiber J. Solano-Iguaran**[9], **Andrew A. Cunningham** [1,11] **& Leonardo D. Bacigalupe** [10,11]

Understanding fine-scale spatial variation in infection risk is central to epidemiology, disease ecology and conservation, yet its causes and consequences remain poorly understood. Here we investigate the dynamics of infection with the aquatic fungus *Batrachochytrium dendrobatidis* (Bd) in several populations of the fully terrestrial Darwin's frog (*Rhinoderma darwinii*) across southern Chile. Using high-resolution spatial capture–recapture data, long-term demographic monitoring and a spatial individual-based model parameterized with empirical estimates, we show that Bd infection in this system exhibits pronounced spatial heterogeneity at scales of only metres. This fine-scale clustering arises from localized transmission of an aquatic pathogen in a terrestrial system, driven by spatial proximity between infected and susceptible individuals. Such transmission generates clustered epidemics and can drive rapid subpopulation collapse in this species, with declines of up to 98% within a year. These epidemics can remain undetected at the broader population level because of spatial decoupling of infection among subpopulations. Our findings provide evidence of epidemic dynamics in a terrestrial Bd host and underscore a broader principle: observational scale fundamentally shapes our ability to detect and interpret infection dynamics in spatially structured populations.

In his seminal paper on the problem of pattern and scale in ecology, Simon A. Levin emphasizes the universal variability of patterns and their underlying mechanisms across spatial scales in natural systems[1]. Unsurprisingly, examples of multiscale spatial patterns of parasite infection are common[2]. For instance, *Batrachochytrium dendrobatidis* (hereafter, Bd)—a pathogen that is one of the primary drivers of contemporary amphibian declines and extinctions and is responsible for the greatest recorded loss of biodiversity attributable to a pathogen[3,4]—has been detected on every continent where amphibians occur[5], yet its spatial distribution appears heterogeneous across all spatial scales examined to date. At global and regional scales, Bd occurrence is closely associated with climatic variables[5]. At the national scale in Chile, however, Bd infections show a stronger association with human footprint than with climate[6]. These observations are consistent with ref. 2, who showed that the factors shaping the distribution of Bd and other pathogens, including West Nile virus and *Borrelia burgdorferi*, vary with the spatial scale of observation. Specifically, they found that biotic factors predicted pathogen distributions only at local scales (~$10^2$–$10^3$ km²), while climate and human population density were significant only at larger, regional scales (typically >$10^4$ km²).

Understanding the mechanisms that shape patterns of infection risk across spatial scales is central to epidemiology, disease ecology

and conservation, and critical for managing infectious diseases in natural populations[2,7–9]. Yet, traditional approaches to studying wildlife diseases have often assumed well-mixed host populations, overlooking the spatial structure that characterizes most free-living populations[10]. As a result, we still know relatively little about the patterns and drivers of fine-scale spatial variation in parasite infection among wild hosts[9]. This is problematic, as failing to observe host–parasite systems at fine spatial scales can obscure key drivers of transmission that might otherwise be targeted through management. For clarity, we define fine-scale variation in infection as spatial structure detectable within host populations, in contrast to between-population patterns more commonly examined in studies of infectious disease[9]. This is a notational convenience, as spatial dependence in host–parasite systems probably occurs along a continuum shaped by host, parasite and environmental characteristics[9].

Moreover, when infection risk is spatially clustered, averaging infection metrics across broad areas can introduce aggregation bias—estimates may appear higher than the risk experienced by most individuals as a result of the disproportionate influence of local infection hotspots, as demonstrated in analyses of COVID-19 data in humans[11]. In spatially structured populations, such averaging can also obscure underlying epidemic dynamics—for example, when infections are sufficiently clustered to produce asynchronous or decoupled dynamics among subpopulations and this variation is hidden by aggregation at the population scale.

Here we investigate the fine-scale spatial dynamics of infection in the Darwin's frog (*Rhinoderma darwinii*)–Bd system. This host–parasite system is particularly well suited for studying spatial variation in infection risk at fine scales. *R. darwinii* is a fully terrestrial amphibian that forms spatially structured populations in forested environments, where Bd—whose infective stage is an aquatic zoospore—is probably transmitted via direct contact or localized environmental exposure, as shown experimentally in other amphibian–chytrid systems[12,13]. Given the host's limited vagility (for example, a median annual displacement of 3.64 m in adults from one studied population[14]) and the potential for Bd to behave like a directly transmitted pathogen in terrestrial settings, we hypothesize that infections will exhibit fine-scale spatial clustering in our study system.

Beyond the question of spatial scale, it is also important to highlight that most research on Bd has focused on amphibians that are aquatic during at least part of their life cycle. Using a comprehensive sample of field studies that examined the impacts of Bd in free-living amphibian populations, we found that only 8% of the 49 species studied were fully terrestrial (Supplementary Information, 'Lifestyle and Bd research'). Yet fully terrestrial amphibians are not uncommon, with at least 34% of amphibians—equivalent to 2,720 species globally—classified as fully terrestrial (either direct-developing species or those with terrestrial larval stages) (Supplementary Information, 'Lifestyle and Bd research'). Therefore, here we also seek to advance understanding of Bd dynamics in fully terrestrial hosts.

To capture the breadth of Bd dynamics in *R. darwinii*, we integrate evidence from two complementary field studies and a spatial individual-based model (IBM) parameterized with empirical estimates. The first field study comprises high-resolution spatial capture–recapture data (±10 cm) from two spatially structured populations (Reserva Forestal Contulmo (RFC) and Reserva Biológica Huilo Huilo (HUI)) in the Austral temperate forests of Chile (Fig. 1a). These data allowed us to quantify the impacts of Bd on host survival, uncover fine-scale spatial patterns of infection risk and examine potential drivers of spatial variation in infection, including microclimatic conditions, host proximity and the diversity of bacterial families in the host's skin microbiome. The second field study involves long-term epidemiological and demographic monitoring of two more subpopulations (Parque Tantauco, site 1 (TAN1) and Parque Tantauco, site 2 (TAN2)) before and after Bd invasion, which provided evidence

of epidemic infection dynamics capable of regulating populations of this fully terrestrial host. Finally, the IBM enabled us to assess the relative importance of spatial, demographic and infection parameters in shaping the host–parasite dynamics in this system and to evaluate potential management interventions such as the exclusion of co-occurring amphibians (hereafter referred to as syntopic). Together, these approaches allow us to characterize the fine-scale spatial dynamics of Bd transmission and provide a mechanistic understanding of how diverse amphibian–Bd outcomes can emerge in terrestrial systems, including rapid Bd-driven population declines and extirpations.

## Results

### Impacts of Bd on host survival

Previous observations from both captive and wild populations suggest that *R. darwinii* is highly susceptible to developing fatal chytridiomycosis[15]. To quantify this susceptibility, we developed a Bayesian spatial multistate capture–recapture (sMCR) model to estimate true, rather than apparent, survival probabilities in Bd-infected and uninfected *R. darwinii* individuals in the wild. This approach overcomes a key limitation of traditional (nonspatial) multistate models, in which survival estimates are confounded by permanent emigration from the study area and are therefore considered only 'apparent'. We found markedly elevated mortality rates in Bd-infected *R. darwinii* individuals (Supplementary Fig. 1). At RFC, monthly survival was reduced by 67.5% (95% Bayesian credible interval (CrI) 45.0–86.1%) in infected compared with uninfected juveniles and by 55.8% (credible interval (CrI) 32.7–77.9%) in infected adults. At HUI, the reduction in survival associated with infection was similarly 66.7% (CrI 23.2–95.1%) in juveniles and 63.3% (CrI 7.0–98.6%) in adults. At RFC, we estimated that the overall probability of dying before recovering from infection was 0.912 (CrI 0.772–0.984) in juveniles and 0.854 (CrI 0.646–0.972) in adults (not estimated at HUI owing to sparse data). Our time-to-death estimates indicate that disease-induced mortality occurs rapidly, with 95% of infected juveniles and adults which do not recover expected to die within 3 months after initial infection (Supplementary Information, 'Bayesian spatial multistate capture–recapture model').

### Fine-scale spatial patterns of infection risk

We found that Bd infection exhibits spatial variation across several scales within the study system. At the broadest spatial scale, there was strong evidence (98.3% probability) for a higher mean monthly probability of Bd infection—defined as the probability of transitioning from uninfected in month $t$ to infected in month $t+1$—in RFC (0.048; CrI 0.034–0.064) compared with HUI (0.015; CrI 0.006–0.028).

When analysing our data at a finer spatial scale within each spatially structured population, we observed high heterogeneity in Bd infection across subpopulations separated by only metres (Fig. 1b and Supplementary Tables 1 and 2). In RFC, 51.3% of the detected infections (39 out of 76) in *R. darwinii* over a 2-year period occurred within a single subpopulation (RFC5), despite only 25.1% of the individuals captured at RFC belonging to that subpopulation. In contrast, just 1.3% of infections (1 out of 76) were recorded in a subpopulation located only 32 m away (RFC4), even though this subpopulation comprised 26.5% of the individuals found in RFC. A similar pattern was observed in HUI, where 76.9% of infections (10 out of 13) over a 4-year period occurred within a single subpopulation (HUI3), despite this subpopulation containing only 17.4% of the individuals in the area.

These observations of fine-scale spatial clustering are further supported by a Bayesian analysis which revealed that during certain months, some subpopulations at both RFC and HUI significantly deviated from expectations under a spatially homogeneous Bd infection risk scenario (Supplementary Fig. 2). This represents one of the finest spatial clustering of parasitic infection ever documented in wildlife[9,16].

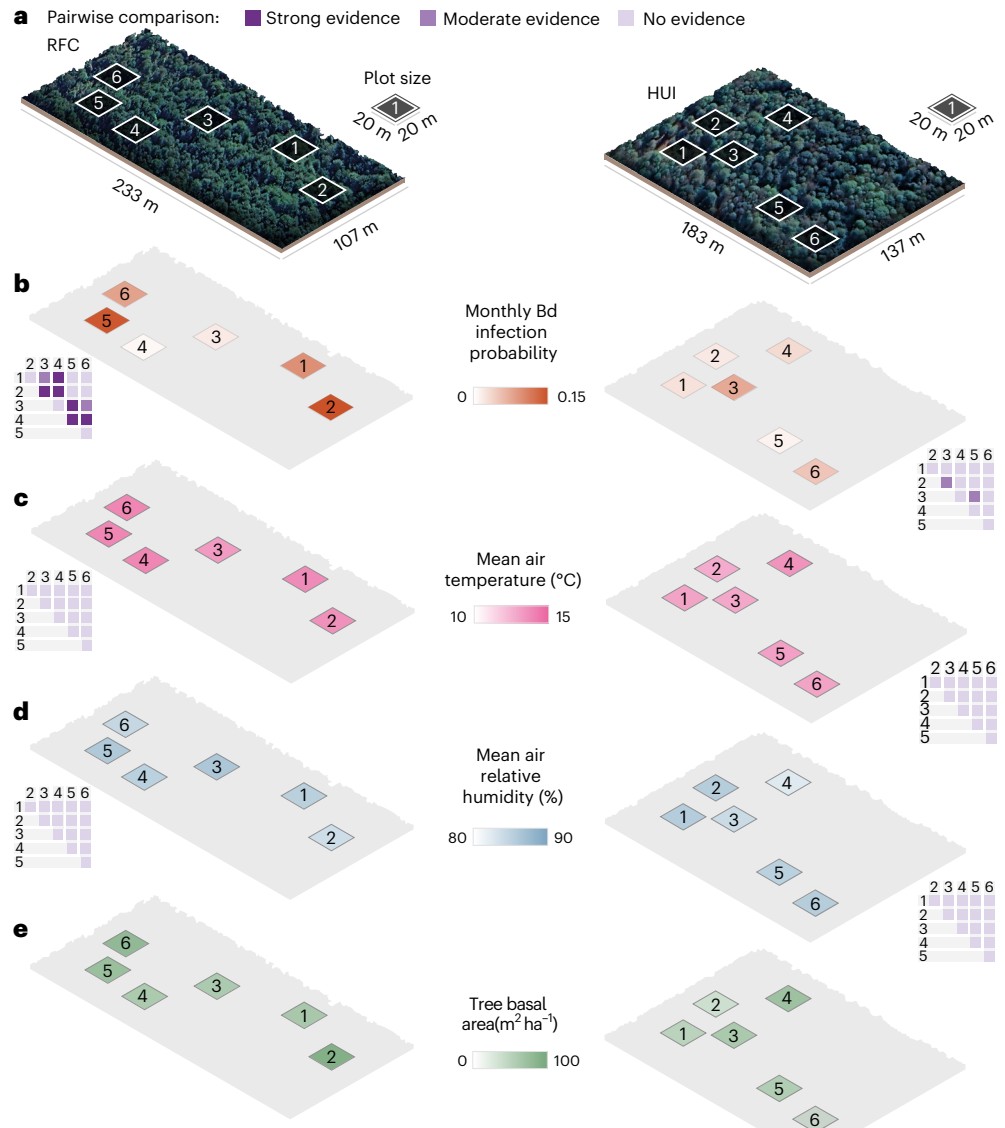

**Fig. 1 | Fine-scale spatial variation in monthly Bd infection probability and environmental variables in two spatially structured populations of *R. darwinii* in southern Chile. a**, Spatial arrangement showing the location of the plots used in the sMCR study. **b**, Mean monthly Bd infection probability in *R. darwinii* frogs estimated using an sMCR model. **c,d**, Mean air temperature (°C) (**c**) and air relative humidity (%) (**d**) measured at 15 cm above ground-level at the centre of each plot. **e**, Tree basal area (m² ha⁻¹) calculated as the sum of the cross-sectional area at the breast height (1.4 m) of all trees having a diameter at the breast height >5 cm. The pairwise comparisons in **b**–**d** indicate the probability that the parameter was different between a pair of populations (exact numbers provided in Supplementary Tables 1–3). Strong evidence, probability ≥0.95; moderate evidence, probability ≥0.80 but <0.95; no evidence, probability <0.80.

## Drivers of fine-scale spatial variation of infection risk

Fine-scale spatial variation in infection prevalence is often thought to result from spatial variation in environmental conditions[16]. However, we found no evidence of differences in temperature- or humidity-related microclimatic variables among the plots inhabited by the studied subpopulations at either RFC or HUI (Fig. 1c,d and Supplementary Table 3; Supplementary Information, 'Environmental variables'). Tree basal area (a proxy of forest biomass that has been linked to among-population variation in *R. darwinii* abundance[17]) varied slightly among plots, but the spatial variation in this variable did not correspond to the spatial variation in Bd infection probability within each area (Fig. 1e and Supplementary Fig. 3). Thus, although climatic factors frequently influence amphibian–Bd host–parasite dynamics[18,19], our results indicate that the fine-scale spatial clustering of Bd infection observed in this system is unlikely to be driven by spatial variation in environmental conditions.

At both RFC and HUI, we found strong evidence (100% probability) that Bd infection probability in *R. darwinii* individuals was positively correlated with the distance to an infected individual in month *t* (Fig. 2b). This relationship was modelled using logistic regression within the sMCR model, where $\alpha_{p_{inf,rd}}$ represents the intercept and $\beta_{p_{inf,rd}}$ the regression slope. At a distance of 0 m, the infection probability was 0.337 (CrI 0.193–0.497) in RFC and 0.645 (CrI 0.143–0.965) in HUI, representing a 7-fold and 43-fold increase in infection probability compared with the area-level mean probabilities, respectively. This finding indicates that intraspecific parasite transmission, whether direct or near-direct (for example, via a contaminated forest floor[12,13]), plays a key role in Bd spread within *R. darwinii* subpopulations and contributes to the fine-scale spatial clustering of infection observed in our system.

We found moderate evidence (85% probability) of a positive relationship between infection probability and the Shannon index

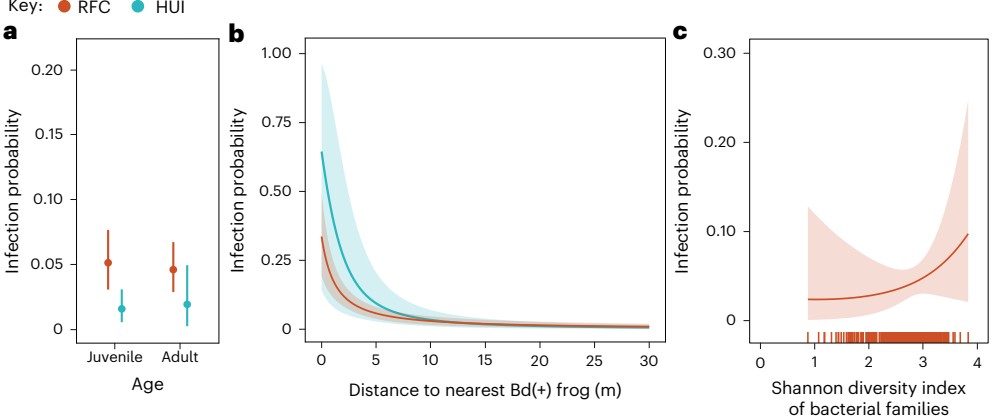

**Fig. 2 | Effect of individual-level variables on monthly Bd infection probability in *R. darwinii* individuals from two spatially structured populations in southern Chile. a**, Bd infection probability (probability of an individual transitioning from uninfected in month *t* to infected in month *t* + 1) in *R. darwinii* juveniles and adults. **b**, Bd infection probability in *R. darwinii* as a function of the distance to the nearest Bd-infected frog in month *t*. **c**, Relationship between the Shannon diversity index of bacterial families on the skin of *R. darwinii* individuals and Bd infection probability. The rug in **c** represents the observed values of the Shannon index. All parameters were estimated using a Bayesian sMCR model. Estimates in **a** and **b** are based on capture–recapture data from 758 *R. darwinii* individuals (RFC, 419; HUI, 339) and estimates in **c** are based on data from 397 *R. darwinii* individuals captured at RFC. Points or lines represent posterior means and error bars or shaded areas indicate 95% Bayesian CrIs.

(alpha diversity) of bacterial families in the skin microbiome of the host (measured only at RFC; Fig. 2c). This may indicate a potential mechanism of host resistance to Bd infection in *R. darwinii*. However, we interpret this result cautiously, as exploration of the raw data revealed that individuals acquiring an infection exhibited only a slightly higher—and statistically nonsignificant—diversity of bacterial families in the preceding months compared with individuals never detected as infected (Supplementary Fig. 4).

### Epidemic Bd infection dynamics in a fully terrestrial host

Earlier studies that averaged Bd infection probability or prevalence across several subpopulations of *R. darwinii* found no evidence of epidemic dynamics and consistently reported low point prevalence[15,20–22]. Consistent with these findings and with the low prevalence observed in other fully terrestrial amphibians, pooled data from subpopulations within each study area in the present study revealed some level of temporal fluctuations in Bd prevalence; however, prevalence never exceeded 16% in either RFC or HUI (Supplementary Fig. 2; Supplementary Information, 'Prevalence and spatial clustering of Bd infection').

Our analyses of high-resolution spatial data demonstrate that epidemic Bd infection events do indeed occur at the subpopulation scale in *R. darwinii*. Using the sMCR model, we estimated monthly variation in Bd infection probability within each subpopulation (Supplementary Fig. 5). This revealed that during the peak of an epidemic event in RFC, up to half the individuals in a subpopulation could acquire Bd infection within a single month (for example, subpopulations RFC1, RFC2 and RFC5; Supplementary Fig. 5).

Our findings of epidemic Bd infection dynamics in subpopulations from RFC and HUI are consistent with observations following the recent invasion of this pathogen into Parque Tantauco, a protected area previously considered a Bd-free refugium and a stronghold for *R. darwinii*[6,21] (Fig. 3). In 2023, after 14 years of epidemiological monitoring in this area, we detected Bd infections for the first time in *R. darwinii* and syntopic amphibians, with an estimated mean Bd prevalence of 25.5% (CrI 14.7%–38.1%) in two *R. darwinii* subpopulations (TAN1 and TAN2). From January 2023 to January 2024, these once-abundant subpopulations, which were near stability before Bd introduction, experienced rapid declines of 91.3% (CrI 82%–96.1%) and 98% (CrI 84.6%–100%), with no signs of recovery as of 2025 (Fig. 3). The TAN2 subpopulation is probably extirpated, as no individuals were detected in 2024 or 2025.

### IBM of Bd infection dynamics in terrestrial systems

To gain a deeper mechanistic understanding of the fine-scale spatial dynamics of Bd infection in our study system and the potential role of syntopic amphibians in these dynamics, we developed a discrete spatial IBM parameterized with empirical estimates (Supplementary Information, 'Individual-based model'). This model describes the spatial, demographic and Bd infection dynamics in a *R. darwinii* subpopulation and a Bd-tolerant syntopic amphibian species over a 12-month period, using monthly time steps. The IBM always starts with the introduction of Bd via a single *R. darwinii* individual infected at month *t* = 1.

A global sensitivity analysis showed that the infection parameters, $\alpha_{p_{\text{inf,rd}}}$ and $\beta_{p_{\text{inf,rd}}}$, consistently exhibited the highest relative importance in shaping the host–parasite dynamics (Supplementary Fig. 6). These parameters define infection probability as a function of the distance to the nearest Bd-infected frog, as described above for the sMCR model. The density of *R. darwinii* ($\lambda_{\text{rd}}$) and syntopic tolerant individuals ($\lambda_{\text{syntopic}}$) also played a notable, albeit less dominant, role in these dynamics. In contrast, other parameters—including survival probabilities of Bd-infected and uninfected frogs, recovery probability from Bd infection, host movement distances and host spatial clustering parameters—had comparatively minor and sometimes negligible, influences on model outputs (Supplementary Fig. 6).

We evaluated plausible combinations of $\lambda_{\text{rd}}$, $\lambda_{\text{syntopic}}$ and infection-parameter values ($\alpha_{p_{\text{inf,rd}}}$ and $\beta_{p_{\text{inf,rd}}}$) that yielded a broad spectrum of host–parasite dynamics. In some scenarios, Bd was unable to invade the *R. darwinii* subpopulation and trigger epidemic events (Supplementary Fig. 7), whereas in others, Bd-driven extirpation of *R. darwinii* occurred within the 12-month period (Supplementary Fig. 8). This range of outcomes could account for the spatial variation in Bd infection probability observed across spatial scales in our host–parasite system. For example, in HUI the empirical mean density of *R. darwinii* was ≤0.05 frogs per m² in all subpopulations (Supplementary Information, 'Individual-based model') and the density of syntopic amphibians was extremely low (only 12 individuals were captured over a 4-year period). At such low values of $\lambda_{\text{rd}}$ and $\lambda_{\text{syntopic}}$, the IBM predicts that Bd will either fail to invade or, if invasion occurs, fade out within the 12-month period, resulting in a very low number of infections (with the proportion of ever-infected individuals at the end of the 12-month period being around 0.2; Supplementary Fig. 39). This prediction is consistent with our field observations of a low prevalence of Bd

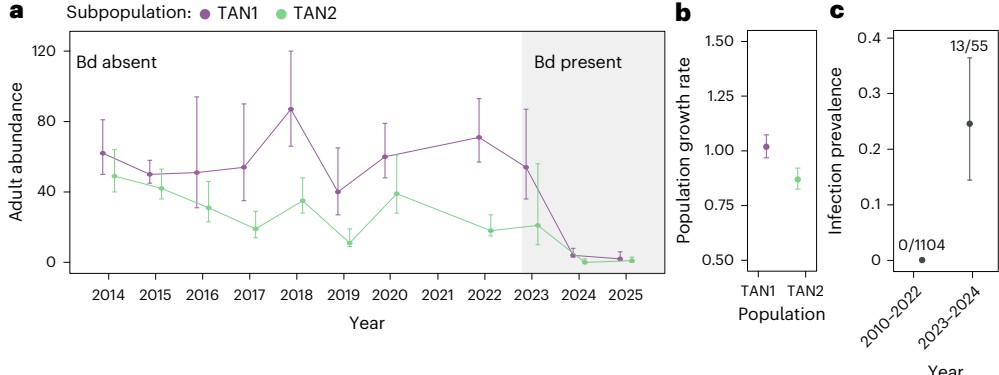

**Fig. 3 | Bd invasion and collapse of two *R. darwinii* subpopulations in Parque Tantauco, Chiloé Island, southern Chile. a**, Adult abundance estimated using a closed capture–recapture model. **b**, Geometric mean population growth from 2014 to 2022 showing that both populations were close to stability (growth rate close to 1) before the first detection of Bd. **c**, Prevalence of Bd infection in *R. darwinii* grouped by the period before the first detection of Bd (2010–2022) and when Bd was present (2023–2024). Data from 2010–2014 also include results from other *R. darwinii* individuals captured close to TAN1 and TAN2 (Supplementary Fig. 9). In 2025, only one *R. darwinii* individual was found,

captured at TAN1, suggesting that TAN2 subpopulation is extinct. In Supplementary Fig. 9c, we show that during the time of Bd invasion and *R. darwinii* population collapse climatic variables in the area were average for the study period. Parameters in **a** and **b** were estimated using a Bayesian closed capture–recapture model fitted to data from 657 *R. darwinii* individuals (TAN1, 418; TAN2, 239). Parameters in **c** were estimated using a Bayesian binomial model. The numbers in **c** represent the number of Bd-positive individuals over the total number of individuals sampled. Points represent posterior means and error bars indicate 95% Bayesian CrIs.

infection in HUI. Similarly, within a spatially structured population, differences in $\lambda_{rd}$ and $\lambda_{syntopic}$ alone can lead to spatial variation in the host–parasite dynamics.

The epidemic ratio in *R. darwinii*, $R_{t,rd}$—defined as the ratio of the number of new infections occurring in month $t + 1$ to the total number of infected individuals present in month $t$ (ref. 23)—exceeded 1 (indicating an epidemic state) during at least 1 month in most simulations, with a maximum of four epidemic months (Supplementary Fig. 7). These results indicate that epidemic events can readily arise in this host–parasite system under parameter values estimated from free-living populations.

Population depression, $\delta$—defined as the relative reduction in the size of the *R. darwinii* subpopulation in the presence of Bd compared with a counterfactual scenario where Bd is absent—was generally high, most notably >0.5 under the mean estimates of the infection parameters from RFC and HUI, with the exception of scenarios where both $\lambda_{rd}$ and $\lambda_{syntopic}$ are very low (Fig. 4 and Supplementary Fig. 8). This indicates that Bd has the capability to drive rapid collapse in wild *R. darwinii* populations, consistent with our observations from TAN1 and TAN2.

A potential disease mitigation action against Bd in *R. darwinii* involves limiting contact between this species and syntopic amphibians through the use of exclusionary fencing and removal of syntopic species[15,24] (Supplementary Information, 'Role of syntopic amphibians in Bd transmission'). To assess the potential efficacy of this approach, we evaluated the relationship between Bd-driven population depression and $\lambda_{syntopic}$. On the basis of the IBM outcomes, this relationship was accurately described by the Gompertz function:

$$\delta(\lambda_{syntopic}) = L \exp(-b \exp(-c\lambda_{syntopic})) \qquad (1)$$

(Fig. 4b), where $L$ is the asymptotic maximum of $\delta$, $b$ controls the horizontal shift of the curve and $c$ determines its rate of increase. Its derivative, $\frac{d\delta}{d\lambda_{syntopic}} = Lbc \exp(-c\lambda_{syntopic}) \exp(-b \exp(-c\lambda_{syntopic}))$, reveals how changes in syntopic-host density translate into shifts in $\delta$. From this analysis, two key management insights emerge. First, the greatest proportional reduction in $\delta$ per syntopic individual removed occurs at low values of $\lambda_{syntopic}$ (Fig. 4c). Although this effect diminishes as $\lambda_{syntopic}$ increases, our simulations indicate that the stabilization

point (beyond which further syntopic-host removal has negligible impact on $\delta$) lies well above the maximum $\lambda_{syntopic}$ range explored in the IBM and was >0.5 frogs m$^{-2}$ in all scenarios. Consequently, even modest reductions in the density of the tolerant syntopic host can still yield measurable benefits in mitigating Bd-driven population depression in *R. darwinii*.

Second, the magnitude of this benefit is highest when *R. darwinii* densities are low (Fig. 4c). As $\lambda_{syntopic}$ grows, however, the difference in impact across varying $\lambda_{rd}$ levels narrows. This arises because, as commonly observed in host–parasite systems[25,26], Bd transmission in our simulated system saturates at higher total-host densities, reducing the marginal gain from removing additional syntopic hosts—particularly if intraspecific transmission within *R. darwinii* is already representing a high proportion of the overall Bd transmission in the system. Taken together, these findings indicate that interventions aimed at lowering syntopic-host density are most potent at lower densities of *R. darwinii* and syntopic tolerant species, highlighting a disease mitigation action that could rescue terrestrial amphibian populations under severe Bd pressure.

## Discussion

Fine-scale spatial variation in infection is increasingly recognized as a widespread feature of host–parasite systems, with important implications for understanding pathogen transmission[9,10]. For example, a long-term pattern of infection clustering in the *Meles meles*–*Mycobacterium bovis* system allowed researchers to identify that social structure is probably a critical determinant of infection risk[10]. Here we show that infection with an aquatic pathogen can display pronounced spatial heterogeneity at very fine scales in spatially structured populations of a fully terrestrial host, with marked differences among subpopulations only metres apart. This clustering was not explained by spatial variation in microclimatic conditions but rather by direct or near-direct host contact. Our empirical results and IBM simulations show that such localized transmission can generate epidemic events and drive rapid subpopulation collapses, with infection parameters and host densities exerting the greatest influence on these dynamics. These findings have important applied implications, as our empirically informed simulations suggest that reducing the density of syntopic tolerant hosts could mitigate Bd-driven population depression in *R. darwinii*.

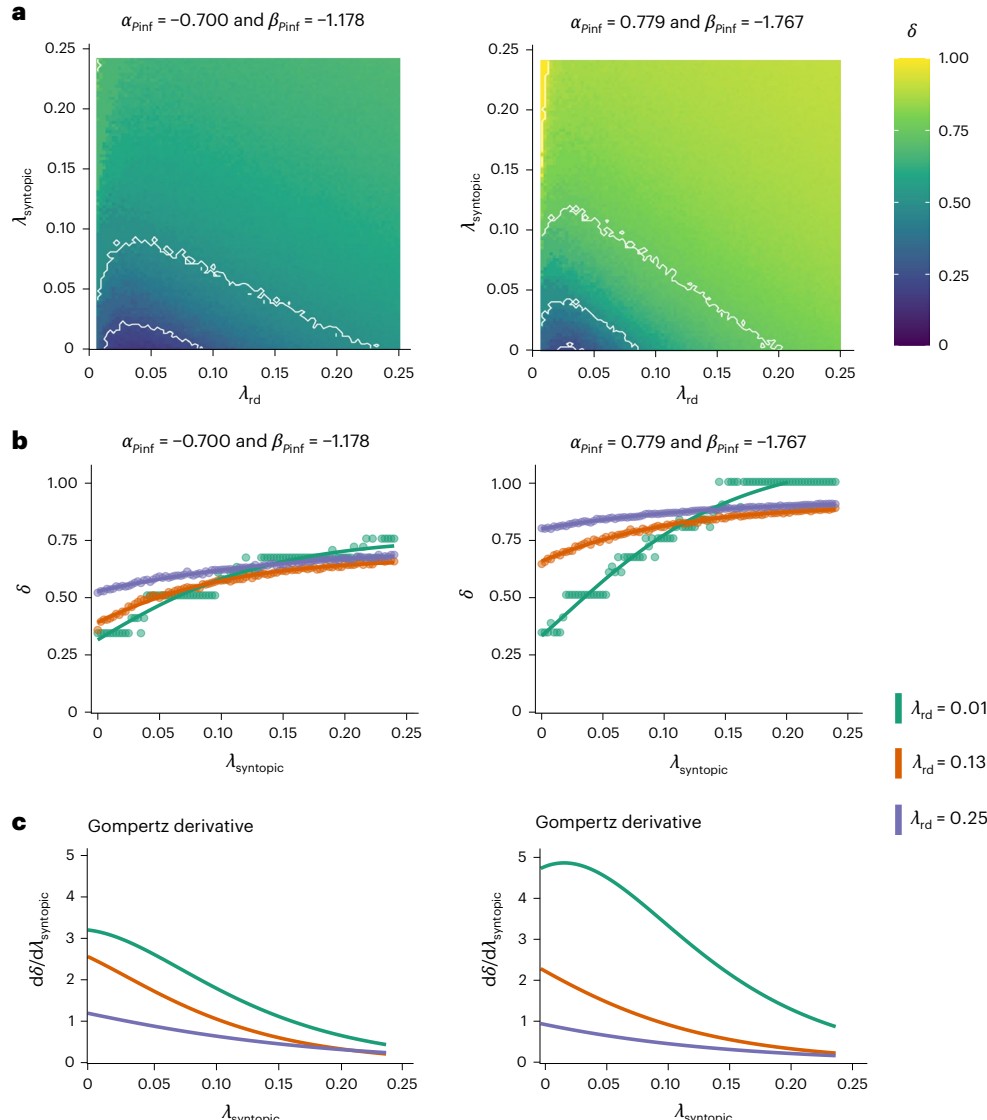

**Fig. 4 | Relationship between host density and Bd-driven population depression ($\delta$) in *R. darwinii* in a 12-month period, predicted by a spatial IBM. a**, The influence of $\lambda_{rd}$ and $\lambda_{syntopic}$, which denote the density (frogs m⁻²) of *R. darwinii* and a tolerant host, respectively, on $\delta$ is shown for two combinations of infection parameters ($\alpha_{P_{inf,rd}}$ and $\beta_{P_{inf,rd}}$), representing the mean estimates of these parameters from RFC and HUI. **b,c**, The relationship between $\delta$ and $\lambda_{syntopic}$, which is well described by the Gompertz function $\delta(\lambda_{syntopic}) = L \exp(-b \exp(-c\lambda_{syntopic}))$ (**b**) and its derivative ($\frac{d\delta}{d\lambda_{syntopic}}$) (which indicates the sensitivity of $\delta$ to changes in $\lambda_{syntopic}$—that is, how much the population depression changes when $\lambda_{syntopic}$ is increased by a small amount), for three values of $\lambda_{rd}$ (**c**). The nonlinear relationship observed in the contour lines in

**a** arises because at very low $\lambda_{rd}$, stochastic infection events cause large relative impacts on $\delta$, leading to high population depression when compared with a no-Bd baseline. As $\lambda_{rd}$ increases, the effect of stochasticity is reduced, resulting in a temporary plateau in depression. However, once the host density is high for the pathogen to invade, sustained Bd transmission drives further population depression, and as $\lambda_{rd}$ increases and Bd transmission approaches saturation a lower value of $\lambda_{syntopic}$ is required to produce an equivalent population depression. The results in **a** and **b** represent the median value from 1,000 simulations for each parameter combination. Contour lines in **a** indicate $\delta$ values of 0.25, 0.5, 0.75 and 1. Value $\delta = 0$ means no depression, while $\delta = 1$ indicates total extinction due to Bd-induced mortality. Intermediate values reflect increasing levels of population depression.

Our findings are also reminiscent of the universal variability of patterns and their underlying mechanisms across spatial scales in natural systems[1]. In our system, low point prevalence—and the resulting limited sample sizes of Bd-infected individuals in previous studies—led us to examine infection as an aggregate process across several local units within spatially structured populations[15,20–22]. At this broader scale, epidemic dynamics appear uncommon, probably because populations able to persist with Bd infection are those in which infection dynamics are decoupled among subpopulations. This decoupling stabilizes average prevalence across the larger spatially structured population, as observed in RFC and HUI. Consequently, in previous studies, we

have concluded that epidemic dynamics are not a prevalent feature of the *R. darwinii*–Bd system[15]. By examining the system at a finer spatial scale, we uncovered clear evidence of epidemic dynamics within *R. darwinii* subpopulations.

Since the discovery of Bd in the mid-1990s, this pathogen has become a model system in wildlife disease research as a result of its exceptionally broad host range and ecological impacts[3,27,28]. However, this body of research has been strongly biased towards amphibians that are aquatic during at least part of their life cycle (hereafter, aquatic species). Importantly, our results highlight that Bd transmission dynamics can differ between aquatic and fully terrestrial hosts, with

important implications for the design of management strategies aimed at reducing transmission or enhancing host population resilience to Bd infection.

First, the fine-scale clustering of Bd infection observed in our system reflects both the limited vagility of the host[14] and the biology of Bd, an aquatic pathogen constrained in terrestrial environments. Our results demonstrate that near-time spatial overlap between infectious and susceptible individuals is necessary for transmission, probably occurring via direct contact or localized environmental exposure[12,13]. In fully terrestrial amphibians, such constraints cause Bd to behave more like a directly transmitted pathogen—where contact rates between infected and susceptible individuals, often correlated with host density, are critical for transmission[27,28], as supported by our spatial IBM. This contrasts with aquatic species, in which environmental transmission from the aquatic zoospore pool can play a central role[29–31] and where host density may play only a minor role in the host–parasite dynamics[18].

Second, a decoupling of Bd infection and demographic dynamics among subpopulations may buffer the impacts of Bd infection at the scale of spatially structured populations[32], as experimentally demonstrated in *Paramecium caudatum* metapopulations affected by the bacterial parasite *Holospora undulata*[33,34]. Given their low vagility and potential low dispersal rates[14,35], potentially low levels of synchrony in spatially structured populations of fully terrestrial amphibians might protect them from Bd-driven extinction—a hypothesis warranting further testing. If confirmed, management strategies aimed at maintaining or reducing synchrony in spatially structured populations might be effective for enhancing host stability and resilience to Bd in this group of amphibians.

Bd prevalence in fully terrestrial amphibians is typically lower than in aquatic species, probably due to reduced exposure to zoospores in terrestrial environments[19,36]. However, low prevalence does not necessarily imply limited population-level impact. If the probability of disease-induced mortality is high—as appears common in Bd-infected fully terrestrial amphibians[15,37]—a pathogen can regulate host populations even at very low prevalence, as predicted by epidemiological theory[15,38]. Owing to potentially reduced transmission rates, the rate of decline may be slower in fully terrestrial species than in aquatic ones, particularly when the density of the focal terrestrial host or syntopic amphibians is moderate to low. Our findings provide strong empirical support that Bd can drive rapid population declines and even extirpation in fully terrestrial hosts, at least at the subpopulation level. Preliminary analyses of monitoring data from Tantauco further suggest that Bd invasion may have caused the extirpation of entire populations within just a few years of pathogen arrival (A.V.-S., unpublished data). These results demonstrate that Bd, present in Chile since the 1970s[20], remains a major threat to *R. darwinii* and continues to drive severe declines decades after its emergence in Chile, highlighting the ongoing threat of this amphibian pandemic. They also suggest that the risk posed by Bd to fully terrestrial amphibians—and thus its global conservation significance—may be critically underappreciated.

More broadly, host–parasite dynamics similar to those observed in our study may be common in spatially structured host populations with limited dispersal, particularly when pathogens are directly transmitted or rely on short-lived, localized environmental transmission. Our results provide rare empirical evidence highlighting the importance of selecting a relevant scale of observation to detect epidemic dynamics and to understand parasite-driven population regulation in wildlife systems.

## Methods

### Study design

This article integrates empirical data derived from two distinct field studies along with one in silico study. Field study 1 involves observational research conducted on two spatially structured populations of *R. darwinii*. Field study 2 encompasses long-term epidemiological and demographic monitoring of two local *R. darwinii* populations. The in silico study complements these empirical observations by modelling the host–parasite dynamics using a spatial IBM. A detailed description of the methods is provided in Supplementary Information.

### Field study 1

From 2018 to 2022, we studied the spatiotemporal dynamics of Bd infection in two spatially structured populations of *R. darwinii* located in two geographical areas 230 km apart in southern Chile: RFC and HUI. The study was concluded prematurely at RFC after March 2020 because of a surge in violent attacks in the area, including an incendiary attack on our research team at the study site. We collected individual-level demographic, search-encounter spatial capture–recapture and Bd infection data on 1,720 captures from 758 *R. darwinii* individuals (RFC, 419; HUI, 339) that were naturally distributed across six nearby study plots within each area. Mean abundance values ranged among plots from 21 (CrI 15–28) to 45 (CrI 38–52) *R. darwinii* frogs in RFC and from 12 (CrI 6–18) to 21 (CrI 15–28) frogs in HUI (Supplementary Table 8). This species is known to form population clusters with individuals showing extremely high site fidelity[14]. The study plots were selected on the basis of a survey conducted at the onset of this study, with plots centred in areas of apparently highest frog density. Opportunistic observations outside the study plots suggest that *R. darwinii* is much less abundant elsewhere. Low rates of dispersal between plots (mean annual dispersal probability, RFC = 0.046 [CrI 0.023–0.075]; HUI = 0.040 [CrI 0.024–0.059]; Supplementary Information, 'Probability of dispersal'), demonstrated that each study plot held a subpopulation of our model species.

Additionally, we collected data on 259 captures from 228 syntopic amphibians (RFC, 216; HUI, 12), including four species (RFC, Contulmo Ground Frog (*Eupsophus contulmoensis*) and Valdivia Ground Frog (*E. vertebralis*); HUI, Grey Wood Frog (*Batrachyla leptopus*) and Rosy Ground Frog (*E. roseus*)) found inside the study plots.

From November to March, each plot was surveyed once per month (that is, primary capture occasion) during three consecutive days (that is, secondary capture occasions), totalling 10 and 19 primary capture occasions in RFC and HUI, respectively. During each secondary capture occasion, two researchers surveyed each plot for 30 min. *R. darwinii* is a diurnal species, so we visually surveyed each site during daylight hours in a manner that covered the entire plot with an apparently equal search effort throughout the site. Details on searching and handling methodology have been extensively described in previous studies[14,15,21]. Briefly, all captured frogs (*R. darwinii* and syntopic amphibians) were measured (snout-to-vent length, SVL), skin-swabbed for Bd infection detection and skin bacteriome characterization (the latter in RFC only) (section 'Bd infection detection' below), photographed for individual recognition using their colouration patterns and released at the exact point of capture. We assigned each captured *R. darwinii* frog to one of two age classes: juvenile (SVL < 24 mm) and adult (SVL ≥ 24 mm). Each study plot was permanently delimited using plastic strings placed every 10 m, allowing us to record the spatial location (*x*- and *y*-coordinates) of each capture with a ±10-cm precision[14,39].

### Field study 2

From 2014 to 2025, we carried out a capture–recapture study of two *R. darwinii* subpopulations in Parque Tantauco (TAN1 and TAN2), Chiloé Island, southern Chile. These populations are within native old-growth forests, where no other anthropogenic stressors besides Bd are known to occur. As above, the capture–recapture data were collected at two nested levels of capture occasions. Each year, we carried out one primary survey period in early summer during the peak of the reproductive season (January–February). During each primary period, we conducted three to four secondary survey occasions, each consisting of a 1-h daily survey per plot on consecutive days. The searching and sampling methodology is the same as that described for the field study 1, with the only distinction that in this area we did not record the spatial

location of captures. In total, we collected data on 952 captures from 657 *R. darwinii* individuals (TAN1, 418; TAN2, 239).

## Bd infection detection

All samples for Bd detection from field studies were obtained from live amphibians using a dry swab (MW100, Medical Wire & Equipment), which was firmly stroked five times over the ventral abdomen and pelvis, each ventral hind limb (femur and tibia) and the plantar surface of each hind foot, for a total of 35 strokes[20]. DNA was extracted from skin swabs using the PrepMan Ultra protocol and Bd DNA was detected with a validated TaqMan quantitative PCR assay targeting the ITS1/5.8S region[40], with extracts diluted 1:10 and supplemented with bovine serum albumin to reduce PCR inhibition[20]. Negative controls and quantitation standards (also serving as positive controls) at 0.1, 1, 10 and 100 zoospore equivalents were run in duplicate on each qPCR plate. Quantitative standards were prepared from a Bd isolate of the global panzootic lineage (ref. IA043). A swab was considered positive if both replicates showed amplification with clear sigmoid curves. In cases of conflicting results (one replicate positive, one negative), the qPCR was repeated in duplicate; if still conflicting, it was repeated a third time in duplicate using a new dilution from the original extraction. If conflict persisted, the swab was classified as negative[20,22]. We assumed that a Bd-positive swab indicated infection of the swabbed animal. At RFC, the number of *R. darwinii* captures and Bd-positive samples per plot were: RFC1 (130/19), RFC2 (99/9), RFC3 (129/5), RFC4 (271/1), RFC5 (240/39) and RFC6 (85/3), totalling 76 positives. At HUI, the corresponding values were: HUI1 (57/0), HUI2 (114/1), HUI3 (142/10), HUI4 (181/2), HUI5 (154/0) and HUI6 (118/0), totalling 13 positives. At Parque Tantauco, the corresponding values were TAN1 (592/9) and TAN2 (360/4), totalling 13 positives.

## Capture–recapture models

To estimate Bd infection and recovery probability, as well as host survival in field study 1, we developed a Bayesian sMCR model (Supplementary Information, 'Bayesian spatial multistate capture–recapture model'). This model considers two states, Bd-infected and uninfected. The return rate of infected frogs (percentage of infected frogs that were recaptured at least once during the course of the study) was very low (only 11 frogs in RFC and one in HUI). Therefore, recovery probability could only be estimated for RFC. Of these 12 frogs, only two individuals (16.7%) with low infection intensities (11 and 12 zoospores) were later recaptured as uninfected only (Supplementary Fig. 12), suggesting infection clearance. In all ten *R. darwinii* individuals detected as infected across 2 months, infection intensity was considerably higher at the later month (Supplementary Fig. 12), indicating infection progression and potential disease development. Owing to the low return rate of infected frogs, the recapture probability of infected frogs was set as equal to that of uninfected frogs. In previous studies in *R. darwinii*, we have used simulations to demonstrate that this assumption does not affect the result of a reduced survival probability in infected frogs[17].

In one of our sMCR model variations, we modelled Bd infection probability as a function of the diversity of bacterial families in the skin microbiome of the host. Briefly, for this analysis, we characterized the skin microbiome of 397 *R. darwinii* individuals captured at RFC using 16S ribosomal RNA gene sequencing of skin swabs. Amplicon data were processed with standard pipelines to infer and classify bacterial amplicon sequence variants, followed by filtering and rarefaction to standardize sequencing depth (Supplementary Information, 'Microbiome'). Owing to the complexity inherent in incorporating time-varying, individual-level covariates into capture–recapture models[41], we averaged each bacteriome metric per individual for analysis. Although we evaluated several diversity metrics at the family level—including observed richness, Shannon diversity index and Pielou's evenness (Supplementary Information, 'Microbiome')—we present results only for the Shannon diversity index, as it proved to be the strongest predictor of infection probability. Given the potential for Bd infection to alter host microbiome composition[42] and that our objective was to assess whether the bacteriome represents a mechanism of resistance to Bd in *R. darwinii*, we restricted analyses involving bacteriome predictors to individuals that were either never detected as infected or were detected as infected only once, with bacteriome averages calculated exclusively from data collected before infection detection.

In another of our sMCR model variations, we used the distance to the nearest Bd-positive individuals as a predictor of *R. darwinii* survival probability. This was used as a time-varying, individual-level covariate, using an imputation submodel to deal with missing data (Supplementary Information, 'Bayesian spatial multistate capture–recapture model').

For field study 2, we estimated *R. darwinii* adult abundance during each primary capture occasion in the two study populations using a Bayesian closed capture–recapture population model as described by ref. 41. We used model $M_0$, where detection probability is modelled as being constant over individuals and over time. We constructed a single Bayesian model, but because of differences in the number of secondary capture occasions each year, we estimated the parameters for each primary occasion separately. In the Bayesian model, we estimated as a derived quantity the annual population growth rate defined as $N_{t+1}/N_t$, where $N_t$ is adult abundance at year $t$. We also calculated as a derived quantity the geometric mean population growth rate for the period 2014–2022, consistent with the period previous to the first detection of Bd infection in the area that occurred in 2023.

All the capture–recapture models were fitted to data in a Bayesian framework with uninformative priors, using JAGS through the R package jagsUI[43–45]. In most cases, we ran four Markov chain Monte Carlo chains with 200,000 iterations, a burn-in of 20,000, without thinning. Chain convergence was evaluated using visual inspection of the chains and the Gelman–Rubin $\hat{R}$ statistic ($\hat{R} < 1.1$)[41].

## Individual-based model

A detailed description of the IBM, including the empirical parameter estimates used to parameterize the model, is provided in Supplementary Information, 'Individual-based model'. A key feature of this IBM is that the spatial location of individuals is simulated using a Log-Gaussian Cox Process model, which consistently and accurately predicted the statistical properties of the spatial distribution of *R. darwinii* frogs in RFC, HUI and TAN2 (Supplementary Information, 'Spatial point pattern analysis'). Each *R. darwinii* individual is assigned a fixed age class (juvenile or adult) and a time-varying infection status (Bd-positive or Bd-negative). Syntopic amphibians are classified only by infection status. All individuals begin the simulation uninfected; to initiate Bd introduction, one randomly selected *R. darwinii* individual is reassigned to an infected state at $t = 1$. At each monthly time step, individuals move and then transition between states. A key feature of the model is that the infection probability for individual $i$ at time $t$ is determined by $\alpha_{p_{\text{inf,rd}}}$ and $\beta_{p_{\text{inf,rd}}}$, as estimated from the sMCR model.

## Ethics statement

This research was approved by the ethics committees at Universidad Austral de Chile (no. 305/2018), Universidad Andrés Bello (no. 13/2015) and Zoological Society of London (no. WLE709 and no. IOZ222) and was conducted in accordance with Chilean law under permits nos. 5666/2013, 230/2015, 212/2016, 1997/2016, 1695/2018, 6618/2019, 7669/2020, 226/2021, 6488/2021, 7161/2022, 7163/2022, 8157/2023 and 6001/2024 of the Servicio Agrícola y Ganadero de Chile and nos. 04/2018 IX, 10/2018 IX and 10/2020 IX de la Corporación Nacional Forestal de Chile.

## Reporting summary

Further information on research design is available in the Nature Portfolio Reporting Summary linked to this article.

## Data availability

The data for reproducing the analyses from this study are available via Zenodo at https://doi.org/10.5281/zenodo.17148244 (ref. 46).

## Code availability

The code used for data analysis is available via Zenodo at https://doi.org/10.5281/zenodo.17148244 (ref. 46).

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

## Acknowledgements

This study is part of an ongoing long-term monitoring programme focused on this threatened species led by the Chilean nonprofit organization ONG Ranita de Darwin and ZSL (www.ranitadedarwin.org/monitoreo). We thank Fundación Parque Tantauco, Corporación Nacional Forestal (CONAF) and Fundación Huilo Huilo for their support. This research was funded by Zoo Leipzig, Rufford Foundation, Weeden Foundation, Mohamed Bin Zayed Species Conservation Fund, Research England, IUCN SSC Internal grant, Stiftung Artenschutz Amphibian Conservation Fund, National Geographic Society and Wild Lama. A.V.-S. and C.A. were supported by the FONDECYT grant nos. 3180107 and 1211587, respectively. L.D.B. was supported by the FONDECYT grant nos. 1200417 and 1240106.

## Author contributions

A.V.-S. designed the research, with contributions from B.R.S., A.A.C. and L.D.B. A.V.-S., S.D.-O., N.H., B.S., A.A.C. and C.A. collected the data. A.V.-S., H.S. and N.H. performed the laboratory analyses. A.V.-S. developed the statistical models and analysed the data, except for the bacteriome data, which were analysed by J.J.S.-I. S.D.-O. designed the figures. A.V.-S. wrote the paper and Supplementary Information, with input from all other authors.

## Competing interests

The authors declare no competing interests.

## Additional information

**Correspondence and requests for materials** should be addressed to Andrés Valenzuela-Sánchez.

¹Institute of Zoology, Zoological Society of London, London, UK. ²ONG Ranita de Darwin, Maipu, Chile. ³Department of Genetics, Evolution and Environment, University College London, London, UK. ⁴One Health Institute, Life Sciences Faculty, Universidad Andres Bello, Santiago, Chile. ⁵Info fauna karch, Neuchatel, Switzerland. ⁶Institut für Evolutionsbiologie und Umweltwissenschaften, Universität Zürich, Zurich, Switzerland. ⁷Université Marie et Louis Pasteur, CNRS, Chrono-environnement (UMR 6249), Besançon, France. ⁸Molecular Epidemiology and Public Health Laboratory, School of Veterinary Science, Massey University, Palmerston North, New Zealand. ⁹Departamento de Salud Hidrobiológica, Instituto de Fomento Pesquero, Puerto Montt, Chile. ¹⁰Instituto de Ciencias Ambientales y Evolutivas, Universidad Austral de Chile, Valdivia, Chile. ¹¹These authors contributed equally: Andrew A. Cunningham, Leonardo D. Bacigalupe. ✉e-mail: andres.valenzuela-sanchez@ioz.ac.uk

# Reporting Summary

## Statistics

For all statistical analyses, confirm that the following items are present in the figure legend, table legend, main text, or Methods section.

| n/a | Confirmed | |
|---|---|---|
| ☐ | ☒ | The exact sample size (*n*) for each experimental group/condition, given as a discrete number and unit of measurement |
| ☐ | ☒ | A statement on whether measurements were taken from distinct samples or whether the same sample was measured repeatedly |
| ☐ | ☒ | The statistical test(s) used AND whether they are one- or two-sided *Only common tests should be described solely by name; describe more complex techniques in the Methods section.* |
| ☐ | ☒ | A description of all covariates tested |
| ☐ | ☒ | A description of any assumptions or corrections, such as tests of normality and adjustment for multiple comparisons |
| ☐ | ☒ | A full description of the statistical parameters including central tendency (e.g. means) or other basic estimates (e.g. regression coefficient) AND variation (e.g. standard deviation) or associated estimates of uncertainty (e.g. confidence intervals) |
| ☐ | ☒ | For null hypothesis testing, the test statistic (e.g. *F*, *t*, *r*) with confidence intervals, effect sizes, degrees of freedom and *P* value noted *Give P values as exact values whenever suitable.* |
| ☐ | ☒ | For Bayesian analysis, information on the choice of priors and Markov chain Monte Carlo settings |
| ☐ | ☒ | For hierarchical and complex designs, identification of the appropriate level for tests and full reporting of outcomes |
| ☒ | ☐ | Estimates of effect sizes (e.g. Cohen's *d*, Pearson's *r*), indicating how they were calculated |

*Our web collection on statistics for biologists contains articles on many of the points above.*

## Software and code

Policy information about availability of computer code

| | |
|---|---|
| Data collection | R (version 4.4.2) was used for data collection |
| Data analysis | R (version 4.4.2) and JAGS (version 4.3.1) were used for data analysis. The data for reproducing the analyses from this study are available via Zenodo at https://doi.org/10.5281/zenodo.17148244. |

For manuscripts utilizing custom algorithms or software that are central to the research but not yet described in published literature, software must be made available to editors and reviewers. We strongly encourage code deposition in a community repository (e.g. GitHub). See the Nature Portfolio guidelines for submitting code & software for further information.

## Data

Policy information about availability of data

All manuscripts must include a data availability statement. This statement should provide the following information, where applicable:

- Accession codes, unique identifiers, or web links for publicly available datasets
- A description of any restrictions on data availability
- For clinical datasets or third party data, please ensure that the statement adheres to our policy

The code used for data analysis is available via Zenodo at https://doi.org/10.5281/zenodo.17148244.

April 2023

# Research involving human participants, their data, or biological material

Policy information about studies with human participants or human data. See also policy information about sex, gender (identity/presentation), and sexual orientation and race, ethnicity and racism.

| | |
|---|---|
| Reporting on sex and gender | *Use the terms sex (biological attribute) and gender (shaped by social and cultural circumstances) carefully in order to avoid confusing both terms. Indicate if findings apply to only one sex or gender; describe whether sex and gender were considered in study design; whether sex and/or gender was determined based on self-reporting or assigned and methods used.*<br>*Provide in the source data disaggregated sex and gender data, where this information has been collected, and if consent has been obtained for sharing of individual-level data; provide overall numbers in this Reporting Summary. Please state if this information has not been collected.*<br>*Report sex- and gender-based analyses where performed, justify reasons for lack of sex- and gender-based analysis.* |
| Reporting on race, ethnicity, or other socially relevant groupings | *Please specify the socially constructed or socially relevant categorization variable(s) used in your manuscript and explain why they were used. Please note that such variables should not be used as proxies for other socially constructed/relevant variables (for example, race or ethnicity should not be used as a proxy for socioeconomic status).*<br>*Provide clear definitions of the relevant terms used, how they were provided (by the participants/respondents, the researchers, or third parties), and the method(s) used to classify people into the different categories (e.g. self-report, census or administrative data, social media data, etc.)*<br>*Please provide details about how you controlled for confounding variables in your analyses.* |
| Population characteristics | *Describe the covariate-relevant population characteristics of the human research participants (e.g. age, genotypic information, past and current diagnosis and treatment categories). If you filled out the behavioural & social sciences study design questions and have nothing to add here, write "See above."* |
| Recruitment | *Describe how participants were recruited. Outline any potential self-selection bias or other biases that may be present and how these are likely to impact results.* |
| Ethics oversight | *Identify the organization(s) that approved the study protocol.* |

Note that full information on the approval of the study protocol must also be provided in the manuscript.

# Field-specific reporting

Please select the one below that is the best fit for your research. If you are not sure, read the appropriate sections before making your selection.

☐ Life sciences ☐ Behavioural & social sciences ☒ Ecological, evolutionary & environmental sciences

For a reference copy of the document with all sections, see nature.com/documents/nr-reporting-summary-flat.pdf

# Ecological, evolutionary & environmental sciences study design

All studies must disclose on these points even when the disclosure is negative.

| | |
|---|---|
| Study description | This article integrates novel empirical data derived from two distinct field studies and one in-silico study. Field Study 1 comprises observational research on two spatially structured populations of Rhinoderma darwinii located in Contulmo (RFC) and Neltume (HUI), southern Chile. In each of these spatially structured populations, six permanent plots, each covering 400 square metres, were established within the forest to collect spatial capture-recapture data. Field Study 2 involves the long-term epidemiological and demographic monitoring of two local populations (TAN1 and TAN2) of R. darwinii from Inio, Chiloé Island, southern Chile, with permanent study plots measuring 700 and 538 square metres, respectively. |
| Research sample | All amphibians found inside the study plots during visual encounter surveys were captured, this included R. darwinii, Eupsophus contulmoensis, E. roseus, E. vertebralis, and Batrachyla leptopus. |
| Sampling strategy | Capture-recapture methods require to capture all individuals observed during a given searching period. In terms of statistical power, the critical aspect is the number of recaptures per individual, as capture-recapture models can generally provide robust parameter estimates even with small sample sizes (e.g. 10-15 individuals per local population) if return rates are relatively high. For our focal species, R. darwinii, recapture probability is generally moderate to high (between 0.3 and 0.6) with the searching effort and methodology used in our study, allowing us to obtain good estimates for most of the parameters of interest. |
| Data collection | Two researchers surveyed each plot for 30 min to 1 hour per day depending on the site. Rhinoderma darwinii is a diurnal species, so we visually surveyed each site during daylight hours in a manner that covered the entire plot with an apparently equal search effort throughout the site. all captured frogs (R. darwinii and syntopic amphibians) were measured (snout-to-vent length), skin-swabbed for Batrachochytrium dendrobatidis infection detection and skin bacteriome characterisation (the latter in RFC only), photographed for individual recognition using their colouration patterns, and released at the exact point of capture. In RFC and HUI, each study plot was permanently delimited using plastic strings placed every 10 m, allowing us to record the spatial location (x- and y-coordinates) of each capture with a ±10 cm precision. |
| Timing and spatial scale | Field Study 1 was conducted from 2018 to 2022. The study was concluded prematurely at RFC after March 2020 due to a surge in violent attacks in the area, including an incendiary attack on our research team at the study site. From November to March each |

year, each plot was surveyed once per month (i.e., primary capture occasion) during three consecutive days (i.e., secondary capture occasions), totalling 10 and 19 primary capture occasions in RFC and HUI, respectively. Field Study 2 was conducted from 2014 to 2025. As above, the capture-recapture data were collected at two nested levels of capture occasions. Each year, we carried out one primary survey period in early summer during the peak of the reproductive season (January-February). During each primary period, we conducted three to four secondary survey occasions, each consisting of a 1-hour daily survey per plot on consecutive days. Additionally, during 2010, 2011, and 2014 we surveyed amphibians and sampled them for B. dendrobatidis infection in eight additional study plots near to TAN1 and TAN2 using the same methodology. The spatial scale of the study plots is described in 'Study description' above.

| Data exclusions | No data were excluded from the analyses. |
|---|---|
| Reproducibility | To ensure reproducibility of this field study, detailed methodological protocols—including the spatial arrangement of study plots, sampling methodology, data collection procedures, and data analysis scripts—have been provided. All data and code required to replicate our findings are publicly available in a dedicated repository (https://doi.org/10.5281/zenodo.17148244), enabling independent verification and reproducibility of the analyses. |
| Randomization | This study is observational and based on naturally occurring spatially structured populations. No experimental treatments were applied, and hence randomization of individuals or groups was not applicable to this study. |
| Blinding | Given the observational nature of this field study, full blinding was not possible during data collection; however, fieldworkers were unaware of the Bd infection status of the captured animals. For laboratory analyses, personnel had access to sample codes and were generally unaware of the species, age, sex, or other characteristics of the animals during DNA extraction and PCR analysis. |

Did the study involve field work?   ☒ Yes   ☐ No

# Field work, collection and transport

| Field conditions | This study was conducted in the Austral temperate forests of Chile, where air temperatures typically range from approximately 5°C to 25°C. Surveys were not conducted during periods of heavy rain or strong winds, as such conditions reduce amphibian detectability and pose an unacceptable risk to fieldworkers. |
|---|---|
| Location | The coordinates of the study sites are: RFC (38°01'50.9''S 73°12'17.0''W), HUI (39°52'21.4''S 71°54'49.2''W), TAN1 (43°21'36.13"S 74°6'20.87"W), TAN2 (43°21'31.60"S 74°6'41.30"W). |
| Access & import/export | Fieldworkers took special care to minimise habitat disturbance during surveys by walking slowly and avoiding damage to vegetation and substrates. Footwear was always cleaned and disinfected with a 1% Virkon S (10 g/L) solution before entering any study site, even if disinfection had already been carried out at a nearby site earlier that same day. Disinfection of footwear and equipment was consistently performed when moving between study sites located more than 1 km apart; in areas with a high risk of B. dendrobatidis spread, this distance threshold was reduced. Other gear and vehicle tyres were cleaned and disinfected with a 1% Virkon S solution when necessary (e.g., in the presence of mud).This research was conducted in accordance with Chilean law under permits no. 5666/2013, no. 230/2015, no. 212/2016, no. 1997/2016, no. 1695/2018, no. 6618/2019, no. 7669/2020, no. 226/2021, no. 6488/2021, no. 7161/2022, no. 7163/2022, no. 8157/2023, no. 6001/2024 of the Servicio Agrícola y Ganadero de Chile, and no. 04/2018 IX, no. 10/2018 IX, and no. 10/2020 IX of the Corporación Nacional Forestal de Chile. |
| Disturbance | We did not observe any evident disturbance to the habitat or the sampled animals. |

# Reporting for specific materials, systems and methods

We require information from authors about some types of materials, experimental systems and methods used in many studies. Here, indicate whether each material, system or method listed is relevant to your study. If you are not sure if a list item applies to your research, read the appropriate section before selecting a response.

## Materials & experimental systems

| n/a | Involved in the study |
|---|---|
| ☒ | ☐ Antibodies |
| ☒ | ☐ Eukaryotic cell lines |
| ☒ | ☐ Palaeontology and archaeology |
| ☐ | ☒ Animals and other organisms |
| ☒ | ☐ Clinical data |
| ☒ | ☐ Dual use research of concern |
| ☒ | ☐ Plants |

## Methods

| n/a | Involved in the study |
|---|---|
| ☒ | ☐ ChIP-seq |
| ☒ | ☐ Flow cytometry |
| ☒ | ☐ MRI-based neuroimaging |

# Animals and other research organisms

Policy information about studies involving animals; ARRIVE guidelines recommended for reporting animal research, and Sex and Gender in Research

| | |
|---|---|
| Laboratory animals | This study did not involve laboratory animals. |
| Wild animals | We made 2,672 captures of 1,415 Rhinoderma darwinii individuals (RFC: 419, HUI: 339, TAN: 657). We also made 383 captures from 352 syntopic amphibians (RFC: 216, HUI: 12: TAN: 124), including eight species (RFC: Contulmo Ground Frog [Eupsophus contulmoensis] and Valdivia Ground Frog [Eupsophus vertebralis]; HUI: Grey Wood Frog [Batrachyla leptopus] and Rosy Ground Frog [Eupsophus roseus]; TAN: Grey Wood Frog [Batrachyla leptopus], Banded Wood Frog [Batrachyla taeniata], Emerald Forest Frog [Hylorina sylvatica], Chiloe Island Ground Frog [Eupsophus calcaratus] and Emilio's Ground Frog [Eupsophus emiliopugini]) found inside the study plots. All post-metamorphic amphibians found during the searches were captured, without differentiating by sex. |

These procedures applied to all amphibian species captured in this study.

Source of animals / capture techniques:
All amphibians were studied in the field using visual encounter surveys in plots. During each survey period, each site was generally surveyed daily on three or four consecutive days by two researchers, in a manner that covered the entire plot with an apparently equal search effort throughout the site. Any frogs seen were captured by hand while wearing a new pair of disposable, powder-free nitrile gloves. Captured frogs were maintained individually in clean, disposable plastic bags filled with air and kept out of direct sunlight until they were processed. Only post-metamorphic individuals were captured.

Handling:
Each amphibian was always handled using clean, disposable, powder-free nitrile gloves. A pair of gloves was used to handle only one individual and was then safely disposed of.

Sampling:
Sampling consisted of the procedures detailed below. Importantly, sampling lasted no more than five minutes per individual, and animals were always handled while wearing a new pair of clean, powder-free nitrile gloves.

Morphometric measures and identification: Captured animals were measured (snout-to-vent length, SVL) using digital callipers, weighed using a digital scale, and photographed for individual identification using unique colouration patterns.

B. dendrobatidis detection from amphibian skin: We used a sterile, dry, rayon-tipped swab (MW100, Medical & Wire Equipment Co.) to sample for B. dendrobatidis DNA that may have been present on the skin of captured frogs. For this, a new swab was firmly run five times each over the ventral abdomen and pelvis, each ventral hind limb (femur and tibia), and the plantar surface of each hind foot, to complete a total of 35 strokes per individual. Using a second swab, this process was immediately repeated in some individuals to estimate pathogen detectability. Each frog was sampled for B. dendrobatidis detection no more than once per month (either using a single swab or two swabs in tandem), up to five times per year.

Release:
After sampling, each amphibian was released at the exact point of capture, which had been marked using a coloured peg. Release always took place within 3 hours of capture.

| | |
|---|---|
| Reporting on sex | Sex-specific analyses were not conducted in this study, and sampling was not biased by sex. |
| Field-collected samples | Skin swabs were stored at ambient temperature in a dry, cool place in the field, and stored at –20°C in the lab until processing. |
| Ethics oversight | This research was approved by the ethics committees at Universidad Austral de Chile (no. 305/2018), Universidad Andrés Bello (no. 13/2015), and Zoological Society of London (no. WLE709 and no. IOZ222). |

Note that full information on the approval of the study protocol must also be provided in the manuscript.

# Plants

| | |
|---|---|
| Seed stocks | *Report on the source of all seed stocks or other plant material used. If applicable, state the seed stock centre and catalogue number. If plant specimens were collected from the field, describe the collection location, date and sampling procedures.* |
| Novel plant genotypes | *Describe the methods by which all novel plant genotypes were produced. This includes those generated by transgenic approaches, gene editing, chemical/radiation-based mutagenesis and hybridization. For transgenic lines, describe the transformation method, the number of independent lines analyzed and the generation upon which experiments were performed. For gene-edited lines, describe the editor used, the endogenous sequence targeted for editing, the targeting guide RNA sequence (if applicable) and how the editor was applied.* |
| Authentication | *Describe any authentication procedures for each seed stock used or novel genotype generated. Describe any experiments used to assess the effect of a mutation and, where applicable, how potential secondary effects (e.g. second site T-DNA insertions, mosiacism, off-target gene editing) were examined.* |

