## [Peer Review File · Nature Ecology & Evolution]

Localized transmission of an aquatic pathogen drives hidden epidemics and population collapse in a terrestrial host

Corresponding Author: Dr Andres Valenzuela-Sanchez

Version 0:

Decision Letter:

21st July 2025

Dear Dr Valenzuela-Sanchez,

Your manuscript entitled "Localized transmission of an aquatic pathogen drives hidden epidemics and collapse in a terrestrial host" has now been seen by 3 reviewers, whose comments are attached. The reviewers have raised a number of concerns which will need to be addressed before we can offer publication in Nature Ecology & Evolution. We will therefore need to see your responses to these points, along with a revised manuscript, before we can reach a final decision regarding publication.

From an editorial perspective, we would also recommend you consider some revisions to the first paragraph. We felt that this very general introduction, and the focus on measles, rather leads the reader down a path that the rest of the paper does not follow. Perhaps this could be shortened or removed.

We therefore invite you to revise your manuscript taking into account all reviewer and editor comments. Please highlight all changes in the manuscript text file.

* If you have not done so already please begin to revise your manuscript so that it conforms to our Article format instructions at <http://www.nature.com/natecolevol/info/final-submission>. Refer also to any guidelines provided in this letter.

* Extended Data Figures - please ensure that any supplementary figures and tables that are crucial to the manuscript's conclusions are converted into Extended Data figures and tables to increase visibility of these data. Extended Data figures and tables are online-only (present in the online PDF and full-text HTML versions of the paper), peer-reviewed display items that provide essential background to the article but are not included in the main article due to space constraints. A maximum of ten Extended Data display items (figures and tables) is permitted.

Link Redacted

Nature Ecology & Evolution is committed to improving transparency in authorship. As part of our efforts in this direction, we are now requesting that all authors identified as 'corresponding author' on published papers create and link their Open Researcher and Contributor Identifier (ORCID) with their account on the Manuscript Tracking System (MTS), prior to acceptance. ORCID helps the scientific community achieve unambiguous attribution of all scholarly contributions. You can create and link your ORCID from the home page of the MTS by clicking on 'Modify my Springer Nature account'. For more information please visit www.springernature.com/orcid.

[redacted]

Reviewer comments:

Reviewer #1 (Remarks to the Author):

Review of "Localized transmission of an aquatic pathogen drives hidden epidemics and collapse in a terrestrial host" for Nature Ecology and Evolution

Thank you for the opportunity to review this paper. As the authors state, the empirical research reported here is indeed novel and makes an important contribution to advancing our understanding of wildlife diseases, and more specifically, chytrid–amphibian dynamics. In particular, the fine-scale dynamics of chytrid infections in terrestrial amphibian species represents a major knowledge gap.

I do not have any concerns about the rigour of the research; the data collection and analyses are excellent. My comments are mostly minor, and the more substantive ones primarily relate to ensuring that the full scope of work—as presented in the supplementary material—is appropriately reflected in the main text. Most readers will only look at the main text, and it was not until I had read through the supplementary material that I fully appreciated the scope of the work the authors have completed. I realise that the main text has word count limitations, but I believe there are opportunities to better communicate the breadth of the study.

The supplementary material is very detailed. While the volume of content may be difficult for readers to digest, I support the approach of providing all the information and enabling readers to explore in detail the methods and supplementary results if they are interested in particular elements.

I also have some suggestions regarding elements of the results/discussion that could be further emphasised.

Overall, the authors should be congratulated for assembling robust datasets to thoroughly investigate an important and understudied aspect of chytrid–amphibian dynamics—particularly in light of the seemingly challenging field conditions described on L341.

Specific Comments:

L26: Suggest replacing 'vulnerable' with 'susceptible', which may be more appropriate in this context.

L31: Consider removing the reference to the absence of an association between environmental conditions and chytrid—it may improve clarity.

L40: While I appreciate the reference to Levin, the discussion of measles in humans may not be the best use of limited word count. Consider focusing on wildlife diseases from the outset and providing examples of contrasting patterns at different spatial scales. In the second paragraph, it is appropriate to narrow the focus to *Batrachochytrium dendrobatidis*, but more detail on macroecological correlates of Bd would be helpful. Then narrow to landscape-scale patterns (1 to 10s of kilometres), where a substantial body of research exists. This would set up the knowledge gap—lack of fine-scale understanding—nicely before transitioning to the paragraph starting on L57.

L82: This paragraph does not fully convey the scope of the work. Consider reworking it to include a numbered list aligned with the results section subheadings. For example, mention the work on skin bacteria, the management experiment excluding co-occurring species, and the simulations. These elements currently appear unexpectedly later in the manuscript.

L235: While $\lambda_{\text{syntopic}}$ is introduced earlier, the term 'syntopic species' appears only here for the first time. For a general audience, consider using 'co-occurring amphibians'. Including species names and relative densities (I know some details are in the SM) would help contextualise their potential role in Bd dynamics.

L262: The statement 'Fine-scale spatial variation in infection is increasingly recognized' would benefit from examples. Provide context—what disease systems? How has this understanding improved broader disease ecology/understanding in that system?

L264: A brief recap of the core findings here would help set up the discussion more effectively.

L266: The discussion jumps straight into system-specific findings. Consider reordering to begin with broader implications, then narrow to the study species/system.

L280–288: This is a valuable part of the research, but it appears unexpectedly. I understand the structure of NEE articles places methods after the discussion, but better signposting in the introduction would help readers grasp the full scope earlier.

L309 and L178: The ongoing severe declines due to Bd are sad to read about. I have observed similar patterns in my own work. In the broader literature, there is a perception that Bd has already done its damage. If they believe it appropriate, the

authors might emphasise that Bd continues to drive declines decades after its emergence and remains a major conservation threat.

L303–308: This is an interesting and well-supported idea. It contrasts with findings from a recent study on a semi-aquatic amphibian, where connectivity was shown to support persistence despite Bd impacts (see: Age truncation due to disease shrinks metapopulation viability for amphibians, *Journal of Animal Ecology*, <https://doi.org/10.1111/1365-2656.14177>).

Paragraph starting L309: This paragraph makes several important points. One additional consideration: the temporal trajectory of population declines may differ between aquatic and terrestrial species, given differences in transmission rates. The prevailing view in the literature seems to be that Bd has already spread to most suitable areas and that the threat has diminished. However, this study suggests that substantial declines can still occur decades later, and that Bd may continue to spread regionally even in countries where it has long been present. Importantly, the temporal scales across which declines occur may be considerably greater in terrestrial amphibian species. I think this is an important point to emphasise.

L344 and Figure 1: It would be helpful to clarify whether the species was largely absent from areas between the 20 × 20 m subplots at RFC and HUI. I presume these quadrants were selected because that is where the frogs occurred and they don't occur in the areas between quadrants.

L398: I suggest starting a new paragraph for the section related to the skin bacteria work. It would also be beneficial to briefly mention in the main text how these samples were processed.

Methods: I recommend including some elements from the detailed methods described in the supplementary material section 'Role of syntopic amphibians in Bd transmission' in the main text. This would help readers better understand the experimental design and rationale.

Figure 1: It would be great to include a photo of one of the plots in this figure to give readers a visual insight into the study system.

L677: Capitalize Bd.

Reviewer #2 (Remarks to the Author):

The authors present a nicely done long-term study on spatial epidemiology using the well-known Bd-amphibian host as the study system. Specifically, the authors studied the critically endangered Darwin's frog, *Rhinoderma darwinii*. Overall, the authors used a combination of capture-recapture data, long-term population monitoring, environmental description, Bd detection and quantification, and individual-base modelling to quantify several epidemiology parameters such as temporal and spatial Bd prevalence and pathogen transmission at fine spatial scale across different sites/populations. The main findings showed that Bd infection is spatially clustered, with high variation between close populations and mostly driven by distance among individuals. This result provided evidence that broad scales studies can mask epidemiologic patterns that occur at the finest scales, which can be very important for an endangered or Bd-vulnerable species management. Overall, I have a few observations (major and minor) that should be easy to address by the authors since they involve reinforcing the authors' findings.

Major concern.

I found the results of this study very helpful for the conservation of the Darwin's frog, especially for those subpopulations that seem to be more vulnerable to Bd infection and potential die-offs. I found the paper robust and convincing in terms of methods and statistical analysis; however I still consider it can add more on the conservation and management for this species. I specifically recommend adding a few more lines at the end of the introduction and discussion (see below), but this topic can be expanded throughout the manuscript.

Paragraph starting on line 380. I found this paragraph very uninformative in terms of replication. Although the authors correctly present relevant references for specific protocols, the manuscript will be highly improved if more details are added in terms of frog handling, Bd extraction, detection and quantification, Bd standards, use or not of internal positive controls, etc., instead of just sending the reader to consult other papers. The citations should remain in the paper but a more explicative background should be added to the benefit of the reader.

Line 57. In this paragraph I recommend adding the following citations

<https://doi.org/10.1073/pnas.1521657113> and <https://doi.org/10.1016/j.gecco.2022.e02197>

These two studies quantified Bd at multiple scales and found similar results using different approaches, which is relevant and provides support for the findings of this study.

Line 72. Add family and common name of your study species. Do the same for the four amphibian species mentioned in line 350-351.

Line 91. Add a last sentence briefly stating the potential applications of this study for the conservation of *R. darwinii*. Same in the last sentence of the discussion (line 326).

Line 361: Please briefly mention that the protocols and materials used for skin swabbing are presented later (section 'Bd infection detection')

Fig 1. I find this figure very appealing and helpful. Add to the caption the meaning of RFC and HUI. Do the same for the other figures where these two sites/populations are being compared. Same observation with TAN1 and TAN2.

Reviewer #2 (Remarks on code availability):

I was able to install and run some of the scripts. The code is a bit messy (clumped lines) but works well and provides reproducibility. It provides a readme file.

Reviewer #3 (Remarks to the Author):

Review of “Localized transmission of an aquatic pathogen drives hidden epidemics and collapse in a terrestrial host” submitted for publication in *Nature Ecology & Evolution*:

The authors integrated empirical data from two field studies with sensitivity/simulation modeling to evaluate the demographic impacts and spatial variation in infection rates of *Batrachochytrium dendrobatidis* (Bd)—the causative agent of chytridiomycosis—between and within populations of Darwin’s frogs (*Rhinoderma darwinii*) that inhabit the temperate forests of Chile. The first field study collected spatial capture-recapture data during 2018–2022 from twelve plots (i.e., subpopulations) nested within two populations (six plots within each population). The second field study, conducted during 2014–2025 at two additional locations, similarly collected capture-recapture data but did not the spatial location of captures. With these data, the authors (1) assessed the impacts of Bd on host survival, (2) quantified spatial variation in infection rates between populations and among subpopulations within each population, (3) examined the potential drivers of such variation in infection rates, (4) documented epizootics after recent invasion of Bd that likely resulted in the near or complete extirpation of subpopulations, and (5) parameterized individual-based models (IBMs) with empirical estimates to simulate Bd infection dynamics, which shed light on the probable efficacy of various management strategies. Given that Bd’s infective stage is an aquatic zoospore, the high mortality in Bd-infected individuals and disease-induced extirpations found in this study are particularly noteworthy because *R. darwinii* is a fully terrestrial species. Further, such fine-scale detail on the spatial distribution of hosts collected in this study revealed localized epizootics within subpopulations that may otherwise have gone undetected if data were aggregated at the population level. Indeed, such findings underscore the importance of selecting a relevant scale of observation to detect epizootic dynamics in wildlife disease systems, with implications for species conservation in the face of emerging pathogens.

The manuscript is well written and timely, as emerging infectious diseases pose a significant threat to global biodiversity, including the effect of chytridiomycosis on amphibians across the world. Despite the severity of this threat, there is somewhat of a paucity of research that examines disease dynamics at multiple scales, though doing so would help to forecast the effectiveness of approaches to mitigation. The multi-scale analysis in this study, along with the funding that spatial proximity between susceptible and infected hosts is key to Bd transmission in terrestrial systems, combine to produce relatively novel contributions to the ecological literature. Further, I believe the authors employed proper and robust methodologies (study design and analytical approach) to test their hypotheses, and provide a thorough supplement and code for context and reproducibility. For all of these reasons, I believe this manuscript would be an excellent contribution to *Nature Ecology & Evolution* and would be of broad interest to the journal’s audience.

I recommend some revisions, detailed below. I feel that these comments should be addressed prior to publication. I hope my suggestions below help the authors improve the clarity of their manuscript.

Major Comments

Reference to “fine-scale” variation in infection rates seems to be a bit overstated throughout the manuscript. In absolute terms, yes, the variation is on the scale of meters. On relative terms, however, the vagility of the focal species would suggest that several meters may not be very fine-scale. For instance, if adults only move ~4 meters (median annual displacement from the manuscript), then 32 meters between subpopulations is relatively far (8x the median distance moved). In some cases, “fine-scale” is used when simply discussing spatial variation in infection rates among subpopulations. In these instances, the novelty and fine-scale language should be toned down a bit, as spatial variation in infection rates among subpopulations is a common finding in the literature. The authors address this briefly in the Discussion (e.g., lines 290–292), though some rephrasing throughout is warranted. With that said, Figure 2b does indicate some extremely fine-scale transmission dynamics with regard to distance of a susceptible host to an infected individual, which I believe highlights the novelty of this work. Thus, when discussing “fine-scale” patterns throughout the manuscript, I would limit the usage of fine-scale to instances of spatial proximity of hosts, rather than variation in overall infection rates of subpopulations.

The authors do a nice job exploring the potential drivers of spatial variation in infection rates among subpopulations. Although environmental conditions (e.g., temperature, moisture) during the study did not influence variation in infection rates, antecedent conditions should not be ruled out, as there often are lag effects with respect to the influence of climate on disease dynamics in wildlife. Some text on this possibility could be added to the Discussion. Further, stochasticity receives little to no attention as a potential driver of spatial variation in infection rates. There are relatively small sample sizes in the study, and the IBM starts in simulation with only one individual becoming infected with Bd. This individual then infects others, which can lead to high infection rates. Given this, the stochasticity involved with respect to one individual becoming infected could play a large role in the spatial variation in infection rates among subpopulations. Thus, more discussion of the role of randomness or chance events in infection dynamics is warranted in the main text.

The authors often focus reporting of infection dynamics as a percentage of detected infections (e.g., Line 120). Given the role of spatial proximity in infection dynamics, it would be beneficial for the reader to have a better understanding of both overall subpopulation abundance and pathogen prevalence (number of Bd positive / total number tested), rather than data on the infections only. Indeed, reporting on the abundance estimates and pathogen prevalence results within the main text would result in a more complete manuscript for the reader.

Minor Comments

When referring to disease outbreaks in wildlife, epizootic often is the preferred term in the ecological literature, rather than epidemic. Consider changing epidemic to epizootic throughout the manuscript.

I wonder how the methodology of this study could be applied to learn about fine-scale patterns of Bd infection dynamics in aquatic/semi-aquatic amphibians? This would make for an impactful Discussion paragraph and inform future studies aiming to conduct this type of work in various systems.

Thank you for the opportunity to review this manuscript. I hope my comments are helpful to the authors in improving the clarity of their manuscript for readers.

Reviewer #3 (Remarks on code availability):

I found the README file thorough and helpful. As time permitted, I ran through some of the code (some of the main analyses) without any trouble. The scripts were well commented. I was not able to run through every script.

*****END*****

Version 1:

Decision Letter:

15th October 2025

Dear Dr. Valenzuela-Sanchez,

Thank you for submitting your revised manuscript "Localized transmission of an aquatic pathogen drives hidden epidemics and collapse in a terrestrial host" (NATECOLEVOL-25051608A). It has now been seen again by the original reviewers and their comments are below. The reviewers find that the paper has improved in revision, and therefore we'll be happy in principle to publish it in Nature Ecology & Evolution, pending minor revisions to satisfy the reviewers' final requests and to comply with our editorial and formatting guidelines.

If you have not done so already, please ensure that you also email us a completed copy of the Reporting summary :

Reporting summary: https://www.nature.com/documents/nr-reporting-summary.pdf

[redacted]

Reviewer #1 (Remarks to the Author):

I believe the authors have done an excellent job of revising their manuscript and providing detailed point-by-point responses to reviewer comments. My comments are very minor and are only suggestions for the author's consideration.

A few minor points for consideration:

L32: I know Bd is generally considered an 'aquatic pathogen', as it is referred to here. But I think your results indicate that it is not exclusively an aquatic pathogen and perhaps we need to stop referring to it as such.

L84: I really like this change to include this information on terrestriality in amphibian species up front. It helps to better highlight the novelty of the empirical contribution provided by this paper.

L226 (paragraph): This is very interesting. In the systems I work in in Australia, an analogous situation seems to be occurring.

L251: Suggest: "This indicates Bd has the capability to drive rapid collapse in wild *R. darwinii* populations, consistent with our observations from TAN1 and TAN2."

L296: The phrasing is a bit strange here. I suggest: "Our findings reinforce the universal variability of patterns and their underlying mechanisms across spatial scales in natural systems".

L338: Thanks for this addition. Maybe an added detail that could be useful in this new sentence is that slower declines in terrestrial species are perhaps more likely at low densities. Your results show that Bd can clearly drive rapid declines in some circumstances in a terrestrial species. (adding this detail will also help better situate this sentence in the context of the rest of the information presented in this paragraph).

Reviewer #2 (Remarks to the Author):

The authors provided a revised version of their manuscript assessing spatial epidemiology (chytridiomycosis) on the critically endangered Darwin's frog. In this new version the authors have addressed successfully most of my concerns. I find the study suitable for publication. Best regards!

Reviewer #2 (Remarks on code availability):

I reviewed the code during my first review and it worked well.

Reviewer #3 (Remarks to the Author):

I very much appreciate the authors for considering each of my comments and revising their manuscript accordingly. When there was disagreement, I found the authors rebuttals thoughtful and respectful, and I do not feel it necessary to reiterate or push back on any previous suggestions that were not incorporated. Further, I do not have any remaining concerns about the manuscript nor the analyses. I believe the authors have addressed each of my main concerns. As presented, I believe this work on variation in infection dynamics within populations (i.e., fine-scale variation in infection rates) will advance how we think about pathogen transmission moving forward, and the methodology used will be applied broadly in disease ecology.

More specific to the revisions, I felt the authors did a nice job better incorporating key information from the supplement into the main text, which provided necessary context and resulted in a more comprehensive manuscript. I also think the authors did a nice job reframing and focusing the beginning of the Introduction. Overall, it appears that the authors did a thorough job in addressing other reviewer comments in addition to considering each of my recommendations. Overall, I congratulate the authors on a nice study. It was a pleasure to review.

Reviewer #3 (Remarks on code availability):

The README file is helpful and I was able to run much of the code. The methodology is robust and has high potential to be useful to others aiming to adopt some of these analytical approaches in their own work.

Reviewer comments

Reviewer #1

Thank you for the opportunity to review this paper. As the authors state, the empirical research reported here is indeed novel and makes an important contribution to advancing our understanding of wildlife diseases, and more specifically, chytrid–amphibian dynamics. In particular, the fine-scale dynamics of chytrid infections in terrestrial amphibian species represents a major knowledge gap.

I do not have any concerns about the rigour of the research; the data collection and analyses are excellent. My comments are mostly minor, and the more substantive ones primarily relate to ensuring that the full scope of work—as presented in the supplementary material—is appropriately reflected in the main text. Most readers will only look at the main text, and it was not until I had read through the supplementary material that I fully appreciated the scope of the work the authors have completed. I realise that the main text has word count limitations, but I believe there are opportunities to better communicate the breadth of the study.

The supplementary material is very detailed. While the volume of content may be difficult for readers to digest, I support the approach of providing all the information and enabling readers to explore in detail the methods and supplementary results if they are interested in particular elements.

I also have some suggestions regarding elements of the results/discussion that could be further emphasised.

Overall, the authors should be congratulated for assembling robust datasets to thoroughly investigate an important and understudied aspect of chytrid–amphibian dynamics—particularly in light of the seemingly challenging field conditions described on L341.

Response: We are grateful for your constructive and encouraging feedback, which has helped us to improve the manuscript. We sincerely appreciate your support.

Specific Comments:

L26: Suggest replacing ‘vulnerable’ with ‘susceptible’, which may be more appropriate in this context.

Response: Replaced as suggested.

L31: Consider removing the reference to the absence of an association between environmental conditions and chytrid—it may improve clarity.

Response: Thanks for this helpful suggestion. We removed that part, and we believe the flow has indeed improved.

L40: While I appreciate the reference to Levin, the discussion of measles in humans may not be the best use of limited word count. Consider focusing on wildlife diseases from the outset and providing examples of contrasting patterns at different spatial scales. In the second paragraph, it is appropriate to narrow the focus to *Batrachochytrium dendrobatidis*, but more detail on macroecological correlates of Bd would be helpful. Then narrow to landscape-scale patterns (1 to 10s of kilometres), where a substantial body of research exists. This would set up the knowledge gap—lack of fine-scale understanding—nicely before transitioning to the paragraph starting on L57.

Response: We agree that the example on measles did not contribute to our manuscript. We have replaced it with the Bd example from the previous version of the manuscript and expanded the paragraph by including a reference to an additional study that tested macroecological correlates of Bd and two other pathogens across multiple spatial scales. This paragraph now reads as follows (L40–53):

“In his seminal paper on the problem of pattern and scale in ecology, Simon A. Levin emphasizes the universal variability of patterns and their underlying mechanisms across spatial scales in natural systems¹. Unsurprisingly, examples of multiscale spatial patterns of parasite infection are common². For instance, *Batrachochytrium dendrobatidis* (Bd)—a pathogen that is one of the primary drivers of contemporary amphibian declines and extinctions^{3,4}—has been detected on every continent where amphibians occur⁵, yet its spatial distribution appears heterogeneous across all spatial scales examined to date. At global and regional scales, Bd occurrence is closely associated with climatic variables⁵. At the national scale in Chile, however, Bd infections show a stronger association with human footprint than with climate⁶. These observations are consistent with Cohen et al.², who showed that the factors shaping the distribution of Bd and other pathogens, including West Nile virus and *Borrelia burgdorferi*, vary with the spatial scale of observation. Specifically, they found that biotic factors predicted pathogen distributions only at local scales ($\sim 10^2$ – 10^3 km²), whereas climate and human population density were significant only at larger, regional scales (typically $>10^4$ km²).”

L82: This paragraph does not fully convey the scope of the work. Consider reworking it to include a numbered list aligned with the results section subheadings. For example, mention the work on skin bacteria, the management experiment excluding co-occurring species, and the simulations. These elements currently appear unexpectedly later in the manuscript.

Response: Thank you for this constructive and helpful comment. We have reworked the paragraph to incorporate your suggestions, and it now reads as follows: (L93-110):

“To capture the breadth of Bd dynamics in *R. darwinii*, we integrate evidence from two complementary field studies and a spatial individual-based model parameterized with empirical estimates. The first field study comprises high-resolution spatial capture–recapture data (± 10 cm) from two spatially structured populations (RFC and HUI) in the Austral temperate forests of Chile (Fig. 1a). These data allowed us to quantify the impacts of Bd on host survival, uncover fine-scale spatial patterns of infection risk, and examine potential drivers of spatial variation in infection, including microclimatic conditions, host proximity, and the diversity of bacterial families in the host’s skin microbiome. The second field study involves long-term epidemiological and demographic monitoring of two additional subpopulations (TAN1 and TAN2) before and after Bd invasion, which provided evidence of epidemic infection dynamics capable of regulating populations of this fully terrestrial host. Finally, the individual-based model enabled us to assess the relative importance of spatial, demographic, and infection parameters in shaping the host–parasite dynamics in this system,

and to evaluate potential management interventions such as the exclusion of co-occurring amphibians (hereafter referred to as syntopic). Together, these approaches allow us to characterize the fine-scale spatial dynamics of Bd transmission and provide a mechanistic understanding of how diverse amphibian–Bd outcomes can emerge in terrestrial systems, including rapid Bd-driven population declines and extirpations.”

L235: While $\lambda_{\text{syntopic}}$ is introduced earlier, the term ‘syntopic species’ appears only here for the first time. For a general audience, consider using ‘co-occurring amphibians’. Including species names and relative densities (I know some details are in the SM) would help contextualise their potential role in Bd dynamics.

Response: Thanks for this helpful comment. We decided to use the term co-occurring amphibians in the last paragraph of the Introduction (L106-107) and state there that the term syntopic will be used hereafter.

L262: The statement ‘Fine-scale spatial variation in infection is increasingly recognized’ would benefit from examples. Provide context—what disease systems? How has this understanding improved broader disease ecology/understanding in that system?

Response: We included the following example in L286-288:

“For example, a long-term pattern of infection clustering in the *Meles meles*–*Mycobacterium bovis* system allowed researchers to identify that social structure is likely a critical determinant of infection risk¹².”

L264: A brief recap of the core findings here would help set up the discussion more effectively.

Response: We agree that a recap can be helpful for readers here, so this paragraph now reads as (L284-297):

“Fine-scale spatial variation in infection is increasingly recognized as a widespread feature of host–parasite systems, with significant implications for understanding pathogen transmission^{11,12}. For example, a long-term pattern of infection clustering in the *Meles meles*–*Mycobacterium bovis* system allowed researchers to identify that social structure is likely a critical determinant of infection risk¹². Here, we show that infection with an aquatic pathogen can display pronounced spatial heterogeneity at very fine scales in spatially structured populations of a fully terrestrial host, with marked differences among subpopulations only metres apart. This clustering was not explained by spatial variation in microclimatic conditions but rather by direct or near-direct host contact. Our empirical results and IBM simulations show that such localized transmission can generate epidemic events and drive rapid subpopulation collapses, with infection parameters and host densities exerting the greatest influence on these dynamics. These findings have important applied implications, as our empirically informed simulations suggest that reducing the density of syntopic tolerant hosts could mitigate Bd-driven population depression in *R. darwini*.”

L266: The discussion jumps straight into system-specific findings. Consider reordering to begin with broader implications, then narrow to the study species/system.

Response: We appreciate this comment. We have made several modifications to the first paragraph of the Discussion to make it more general and not Bd-specific. The revised paragraph is provided in our response to the previous comment (see above). The second paragraph of the Discussion also highlights implications that extend beyond our specific study system, and we conclude with a wrap-up paragraph that places the main findings in the broader context of disease ecology.

L280–288: This is a valuable part of the research, but it appears unexpectedly. I understand the structure of NEE articles places methods after the discussion, but better signposting in the introduction would help readers grasp the full scope earlier.

Response: Thanks for highlighting this. We decided to move these results to the Introduction and to update the corresponding paragraph in the Discussion. The paragraph in the Introduction now reads as follows (L84-92):

“Beyond the question of spatial scale, it is also important to highlight that most research on Bd has focused on amphibians that are aquatic during at least part of their life cycle. Using a comprehensive sample of field studies that examined the impacts of Bd in free-living amphibian populations, we found that only 8% of the 49 species studied were fully terrestrial (Supplementary Material: Lifestyle and Bd research). Yet fully terrestrial amphibians are not uncommon, with at least 34% of amphibians—equivalent to 2,720 species globally—classified as fully terrestrial (either direct-developing species or those with terrestrial larval stages) (Supplementary Material: Lifestyle and Bd research). Therefore, here we also seek to advance understanding of Bd dynamics in fully terrestrial hosts.”

And in the Discussion we updated the paragraph as follows (L309-315):

“Since the discovery of Bd in the mid-1990s, this pathogen has become a model system in wildlife disease research due to its exceptionally broad host range and ecological impacts^{5,29,30}. However, this body of research has been strongly biased toward amphibians that are aquatic during at least part of their life cycle (hereafter, aquatic species). Importantly, our results highlight that Bd transmission dynamics can differ between aquatic and fully terrestrial hosts, with important implications for the design of management strategies aimed at reducing transmission or enhancing host population resilience to Bd infection.”

L309 and L178: The ongoing severe declines due to Bd are sad to read about. I have observed similar patterns in my own work. In the broader literature, there is a perception that Bd has already done its damage. If they believe it appropriate, the authors might emphasise that Bd continues to drive declines decades after its emergence and remains a major conservation threat.

Response: We believe this change is appropriate. We have updated the text at L345–349 as follows:

“These results demonstrate that Bd, present in Chile since the 1970s²², remains a major threat to *R. darwinii* and continues to drive severe declines decades after its emergence in Chile, highlighting the ongoing threat of this amphibian pandemic. They also suggest the risk posed by Bd to fully terrestrial amphibians—and thus its global conservation significance—may be critically underappreciated.”

L303–308: This is an interesting and well-supported idea. It contrasts with findings from a recent study on a semi-aquatic amphibian, where connectivity was shown to support persistence despite Bd

impacts (see: Age truncation due to disease shrinks metapopulation viability for amphibians, Journal of Animal Ecology, <https://doi.org/10.1111/1365-2656.14177>).

Response: Thank you for sharing this paper. We would just like to add that connectivity remains important, because even with low levels of infection and demographic synchrony among subpopulations some dispersal between subpopulations should be needed for this protective effect to operate in a spatially structured population. We plan to use our IBM to explore this theoretically in future work. We did not add further detail to the text, as we feel it lies beyond the scope of the present study.

Paragraph starting L309: This paragraph makes several important points. One additional consideration: the temporal trajectory of population declines may differ between aquatic and terrestrial species, given differences in transmission rates. The prevailing view in the literature seems to be that Bd has already spread to most suitable areas and that the threat has diminished. However, this study suggests that substantial declines can still occur decades later, and that Bd may continue to spread regionally even in countries where it has long been present. Importantly, the temporal scales across which declines occur may be considerably greater in terrestrial amphibian species. I think this is an important point to emphasise.

Response: We have added more details to this paragraph to emphasise the points raised by the reviewer. It now reads as follows (L335–349):

“Bd prevalence in fully terrestrial amphibians is typically lower than in aquatic species, likely due to reduced exposure to zoospores in terrestrial environments^{19,36}. However, low prevalence does not necessarily imply limited population-level impact. If the probability of disease-induced mortality is high—as appears common in Bd-infected fully terrestrial amphibians^{15,37}—a pathogen can regulate host populations even at very low prevalence, as predicted by epidemiological theory^{15,38}. Due to potentially reduced transmission rates, the rate of decline may be slower in fully terrestrial species than in aquatic ones. Our findings provide strong empirical support that Bd can drive population declines and even extirpation in fully terrestrial hosts, at least at the subpopulation level. Preliminary analyses of monitoring data from Tantauco further suggest that Bd invasion may have caused the extirpation of entire populations within just a few years of pathogen arrival (A. Valenzuela-Sánchez, unpublished data). These results demonstrate that Bd, present in Chile since the 1970s²⁰, remains a major threat to *R. darwinii* and continues to drive severe declines decades after its emergence in Chile, highlighting the ongoing threat of this amphibian pandemic. They also suggest the risk posed by Bd to fully terrestrial amphibians—and thus its global conservation significance—may be critically underappreciated.”

L344 and Figure 1: It would be helpful to clarify whether the species was largely absent from areas between the 20 × 20 m subplots at RFC and HUI. I presume these quadrants were selected because that is where the frogs occurred and they don't occur in the areas between quadrants.

Response: To clarify this, we have added the following paragraph at L377-381:

“This species is known to form population clusters with individuals showing extremely high site fidelity¹⁶. The study plots were selected on the basis of a survey conducted shortly before the onset of this study, with plots centred in areas of apparently highest frog density. Opportunistic observations outside the study plots suggest that *R. darwinii* is much less abundant elsewhere.”

L398: I suggest starting a new paragraph for the section related to the skin bacteria work. It would also be beneficial to briefly mention in the main text how these samples were processed.

Response: We followed this recommendation, and the updated paragraph now reads as follows (L448-463):

“In one of our SMCR model variations, we modelled Bd infection probability as a function of the diversity of bacterial families in the host’s skin microbiome. Briefly, for this analysis we characterized the skin microbiome of 397 *R. darwinii* individuals captured at RFC using 16S rRNA gene sequencing of skin swabs. Amplicon data were processed with standard pipelines to infer and classify bacterial Amplicon Sequence Variants, followed by filtering and rarefaction to standardize sequencing depth (Supplementary Material: Microbiome). Due to the complexity inherent in incorporating time-varying, individual-level covariates into capture-recapture models⁴⁴, we averaged each bacteriome metric per individual for analysis. Although we evaluated several diversity metrics—including Observed Richness, Shannon Diversity Index, and Pielou’s Evenness (Supplementary Material: Microbiome)—we present results only for the Shannon Diversity Index, as it proved to be the strongest predictor of infection probability. Given the potential for Bd infection to alter host microbiome composition⁴⁵, and that our objective was to assess whether the bacteriome represents a mechanism of resistance to Bd in *R. darwinii*, we restricted analyses involving bacteriome predictors to individuals that were either never detected as infected or were detected as infected only once, with bacteriome averages calculated exclusively from data collected prior to infection detection.”

Methods: I recommend including some elements from the detailed methods described in the supplementary material section ‘Role of syntopic amphibians in Bd transmission’ in the main text. This would help readers better understand the experimental design and rationale.

Response: Thank you for this comment. We placed this component in the Supplementary Material because it represents preliminary results with several caveats, which we have detailed there. We chose to include it as supplementary information rather than omit it entirely, as we believe it may still benefit readers with a specific applied interest in this intervention. However, we feel that moving it into the main text would add complexity without contributing directly to our central research questions.

Figure 1: It would be great to include a photo of one of the plots in this figure to give readers a visual insight into the study system.

Response: We attempted to incorporate a photograph of the forest, but the figure already contains many elements (including a satellite image of the forest), and adding more made it appear overloaded and detracted from the main information we wish to convey. For interested readers, however, a photograph of the forest type where this species occurs is provided in the Supplementary Figure 43.

L677: Capitalize Bd.

Response: Thank you for detecting this; it has been fixed.

Reviewer #2

The authors present a nicely done long-term study on spatial epidemiology using the well-known Bd-amphibian host as the study system. Specifically, the authors studied the critically endangered Darwin's frog, *Rhinoderma darwinii*. Overall, the authors used a combination of capture-recapture data, long-term population monitoring, environmental description, Bd detection and quantification, and individual-base modelling to quantify several epidemiology parameters such as temporal and spatial Bd prevalence and pathogen transmission at fine spatial scale across different sites/populations. The main findings showed that Bd infection is spatially clustered, with high variation between close populations and mostly driven by distance among individuals. This result provided evidence that broad scales studies can mask epidemiologic patterns that occur at the finest scales, which can be very important for an endangered or Bd-vulnerable species management. Overall, I have a few observations (major and minor) that should be easy to address by the authors since they involve reinforcing the authors' findings.

Response: Thank you for your positive and constructive comments, which we believe have improved our manuscript. We greatly appreciate your help.

Major concern.

I found the results of this study very helpful for the conservation of the Darwin's frog, especially for those subpopulations that seem to be more vulnerable to Bd infection and potential die-offs. I found the paper robust and convincing in terms of methods and statistical analysis; however I still consider it can add more on the conservation and management for this species. I specifically recommend adding a few more lines at the end of the introduction and discussion (see below), but this topic can be expanded throughout the manuscript.

Response: Thank you for highlighting this important point. We have made several revisions to the manuscript to better emphasise the conservation implications of our study (please see our response to Reviewer #2 below for details).

Paragraph starting on line 380. I found this paragraph very uninformative in terms of replication. Although the authors correctly present relevant references for specific protocols, the manuscript will be highly improved if more details are added in terms of frog handling, Bd extraction, detection and quantification, Bd standards, use or not of internal positive controls, etc., instead of just sending the reader to consult other papers. The citations should remain in the paper but a more explicative background should be added to the benefit of the reader.

Response: We have included additional details as follows (L419-435):

“All samples for Bd detection from field studies were obtained from live amphibians using a dry swab (MW100, Medical Wire & Equipment, UK), which was firmly stroked five times over the ventral abdomen and pelvis, each ventral hind limb (femur and tibia), and the plantar surface of each hind foot, for a total of 35 strokes²⁰. DNA was extracted from skin swabs using the PrepMan Ultra™ protocol, and Bd DNA was detected with a validated TaqMan qPCR assay targeting the ITS1/5.8S region⁴⁰, with extracts diluted 1:10 and supplemented with bovine serum albumin to reduce PCR inhibition²⁰. Negative controls and quantitation standards (also serving as positive controls) at 0.1, 1, 10, and 100 zoospore equivalents were run in duplicate on each qPCR plate. Quantitative standards were prepared from a Bd isolate of the Global Panzootic Lineage (ref. IA043). A swab was considered positive if both replicates showed amplification with clear sigmoid curves. In cases of conflicting results (one replicate positive, one negative), the qPCR was repeated in duplicate; if still conflicting, it was repeated a third time in duplicate using a new dilution from the original extraction. If conflict

persisted, the swab was classified as negative^{20,22}. We assumed that a Bd-positive swab indicated infection of the swabbed animal.”

Line 57. In this paragraph I recommend adding the following citations

<https://doi.org/10.1073/pnas.1521657113> and <https://doi.org/10.1016/j.gecco.2022.e02197>

These two studies quantified Bd at multiple scales and found similar results using different approaches, which is relevant and provides support for the findings of this study.

Response: Thank you for recommending these interesting studies. We have incorporated the first one into the manuscript. Although the second is an important contribution, we feel that adding it to this paragraph would not be beneficial, as it does not seem to focus on the drivers of Bd infection across multiple spatial scales. We have therefore included the first suggested reference both in the paragraph highlighted by the reviewer and in the first paragraph (L48–53):

“These observations are consistent with Cohen et al.², who showed that the factors shaping the distribution of Bd and other pathogens, including West Nile virus and *Borrelia burgdorferi*, vary with the spatial scale of observation. Specifically, they found that biotic factors predicted pathogen distributions only at local scales ($\sim 10^2$ – 10^3 km²), whereas climate and human population density were significant only at larger, regional scales (typically $>10^4$ km²).”

Line 72. Add family and common name of your study species. Do the same for the four amphibian species mentioned in line 350-351.

Response: We have included the common names as recommended by the reviewer. We have not added the family names, as we found this disrupted the flow of the text, and given that this is not a taxonomic study, we feel that including this information would not provide a significant improvement.

Line 91. Add a last sentence briefly stating the potential applications of this study for the conservation of *R. darwinii*. Same in the last sentence of the discussion (line 326).

Response: We have now included a direct reference to the conservation implications at L103–107:

“Finally, the individual-based model enabled us to assess the relative importance of spatial, demographic, and infection parameters in shaping the host–parasite dynamics in this system, and to evaluate potential management interventions such as the exclusion of co-occurring amphibians (hereafter referred to as syntopic).”

Additionally, we have incorporated the potential applications of this study for the conservation of *R. darwinii* into the last sentence of the first paragraph of the Discussion, which now reads as follows (L294–297):

“These findings have important applied implications, as our empirically informed simulations suggest that reducing the density of syntopic tolerant hosts could mitigate Bd-driven population depression in *R. darwinii*.”

More broadly, we have made the conservation implications of our study more explicit in the paragraph at L335–349, which now reads as follows:

“Bd prevalence in fully terrestrial amphibians is typically lower than in aquatic species, likely due to reduced exposure to zoospores in terrestrial environments^{21,38}. However, low prevalence does not

necessarily imply limited population-level impact. If the probability of disease-induced mortality is high—as appears common in Bd-infected fully terrestrial amphibians^{17,39}—a pathogen can regulate host populations even at very low prevalence, as predicted by epidemiological theory^{17,41}. Due to potentially reduced transmission rates, the rate of decline may be slower in fully terrestrial species than in aquatic ones. Our findings provide strong empirical support that Bd can drive population declines and even extirpation in fully terrestrial hosts, at least at the subpopulation level. Preliminary analyses of monitoring data from Tantauco further suggest that Bd invasion may have caused the extirpation of entire populations within just a few years of pathogen arrival (A. Valenzuela-Sánchez, unpublished data). These results demonstrate that Bd, present in Chile since the 1970s²², remains a major threat to *R. darwinii* and continues to drive severe declines decades after its emergence in Chile, highlighting the ongoing threat of this amphibian pandemic. They also suggest the risk posed by Bd to fully terrestrial amphibians—and thus its global conservation significance—may be critically underappreciated.”

Line 361: Please briefly mention that the protocols and materials used for skin swabbing are presented later (section ‘Bd infection detection’)

Response: Added as recommended in L398.

Fig 1. I find this figure very appealing and helpful. Add to the caption the meaning of RFC and HUI. Do the same for the other figures where these two sites/populations are being compared. Same observation with TAN1 and TAN2.

Response: Added as recommended to the legend of each figure.

Reviewer #2 (Remarks on code availability):

I was able to install and run some of the scripts. The code is a bit messy (clumped lines) but works well and provides reproducibility. It provides a readme file.

Response: We appreciate you testing our code and are pleased to know it worked well.

Reviewer #3

Review of “Localized transmission of an aquatic pathogen drives hidden epidemics and collapse in a terrestrial host” submitted for publication in *Nature Ecology & Evolution*:

The authors integrated empirical data from two field studies with sensitivity/simulation modeling to evaluate the demographic impacts and spatial variation in infection rates of *Batrachochytrium dendrobatidis* (Bd)—the causative agent of chytridiomycosis—between and within populations of Darwin’s frogs (*Rhinoderma darwinii*) that inhabit the temperate forests of Chile. The first field study collected spatial capture-recapture data during 2018–2022 from twelve plots (i.e., subpopulations) nested within two populations (six plots within each population). The second field study, conducted during 2014–2025 at two additional locations, similarly collected capture-recapture data but did not the spatial location of captures. With these data, the authors (1) assessed the impacts of Bd on host survival, (2) quantified spatial variation in infection rates between populations and among subpopulations within each population, (3) examined the potential drivers of such variation in infection rates, (4) documented epizootics after recent invasion of Bd that likely resulted in the near or complete extirpation of subpopulations, and (5) parameterized individual-based models (IBMs) with empirical estimates to simulate Bd infection dynamics, which shed light on the probable efficacy of various management strategies. Given that Bd’s infective stage is an aquatic zoospore, the high mortality in Bd-infected individuals and disease-induced extirpations found in this study are particularly noteworthy because *R. darwinii* is a fully terrestrial species. Further, such fine-scale detail on the spatial distribution of hosts collected in this study revealed localized epizootics within subpopulations that may otherwise have gone undetected if data were aggregated at the population level. Indeed, such findings underscore the importance of selecting a relevant scale of observation to detect epizootic dynamics in wildlife disease systems, with implications for species conservation in the face of emerging pathogens.

The manuscript is well written and timely, as emerging infectious diseases pose a significant threat to global biodiversity, including the effect of chytridiomycosis on amphibians across the world. Despite the severity of this threat, there is somewhat of a paucity of research that examines disease dynamics at multiple scales, though doing so would help to forecast the effectiveness of approaches to mitigation. The multi-scale analysis in this study, along with the finding that spatial proximity between susceptible and infected hosts is key to Bd transmission in terrestrial systems, combine to produce relatively novel contributions to the ecological literature. Further, I believe the authors employed proper and robust methodologies (study design and analytical approach) to test their hypotheses, and provide a thorough supplement and code for context and reproducibility. For all of these reasons, I believe this manuscript would be an excellent contribution to *Nature Ecology & Evolution* and would be of broad interest to the journal’s audience.

I recommend some revisions, detailed below. I feel that these comments should be addressed prior to publication. I hope my suggestions below help the authors improve the clarity of their manuscript.

Response: We thank you for your thoughtful and constructive comments, which have strengthened our manuscript. Your input is greatly appreciated.

Major Comments

Reference to “fine-scale” variation in infection rates seems to be a bit overstated throughout the manuscript. In absolute terms, yes, the variation is on the scale of meters. On relative terms, however, the vagility of the focal species would suggest that several meters may not be very fine-scale. For instance, if adults only move ~4 meters (median annual displacement from the manuscript), then 32 meters between subpopulations is relatively far (8x the median distance moved). In some cases, “fine-scale” is used when simply discussing spatial variation in infection rates among subpopulations. In these instances, the novelty and fine-scale language should be toned down a bit, as spatial variation in infection rates among subpopulations is a common finding in the literature. The authors address this briefly in the Discussion (e.g., lines 290–292), though some rephrasing throughout is warranted. With that said, Figure 2b does indicate some extremely fine-scale transmission dynamics with regard to distance of a susceptible host to an infected individual, which I believe highlights the novelty of this work. Thus, when discussing “fine-scale” patterns throughout the manuscript, I would limit the usage of fine-scale to instances of spatial proximity of hosts, rather than variation in overall infection rates of subpopulations.

Response: Thank you for this important comment. We recognise that our use of the term “fine-scale” was not sufficiently clear in the previous version, and as this is a relative term, we understand the reviewer’s concern. We have therefore provided a working definition at L61-66:

“For clarity, we define fine-scale variation in infection as spatial structure detectable within host populations, in contrast to between-population patterns more commonly examined in studies of infectious disease¹¹. This is a notational convenience, as spatial dependence in host–parasite systems likely occurs along a continuum shaped by host, parasite, and environmental characteristics (see Albery et al.¹¹ for further discussion).”

We believe this definition is also consistent with previous usage of the term—for example, by Gibson et al. (2016), where *fine-scale* is used to refer to “scales where gene flow between sites is high”, which aligns with among-subpopulation variation.

To our knowledge, most studies exploring spatial variation in infection risk have focused on spatial units that could be considered populations rather than subpopulations. This is precisely the situation in our study, as described in L298–308:

“Our findings are also reminiscent of the universal variability of patterns and their underlying mechanisms across spatial scales in natural systems¹. In our system, low point prevalence—and the resulting limited sample sizes of Bd-infected individuals in previous studies—led us to examine infection as an aggregate process across multiple local units within spatially structured populations^{17, 22–24}. At this broader scale, epidemic dynamics appear uncommon, likely because populations able to persist with Bd infection are those in which infection dynamics are decoupled among subpopulations. This decoupling stabilizes average prevalence across the larger spatially structured population, as observed in RFC and HUI. Consequently, in previous studies we have concluded that epidemic dynamics are not a prevalent feature of the *R. darwinii*-Bd system¹⁷. By examining the system at a finer spatial scale, we uncovered clear evidence of epidemic dynamics within *R. darwinii* subpopulations.”

For the reasons outlined above, we believe it is useful in our article to maintain the definition of fine-scale at the level of among-subpopulation variation, with the considerations stated in L316–318:

“First, the fine-scale clustering of Bd infection observed in our system reflects both the host’s limited vagility¹⁶ and the biology of Bd, an aquatic pathogen constrained in terrestrial environments.”

References

Gibson, A. K., Jokela, J., & Lively, C. M. (2016). Fine-scale spatial covariation between infection prevalence and susceptibility in a natural population. *The American Naturalist*, 188(1), 1–14.
<https://doi.org/10.1086/686767>

The authors do a nice job exploring the potential drivers of spatial variation in infection rates among subpopulations. Although environmental conditions (e.g., temperature, moisture) during the study did not influence variation in infection rates, antecedent conditions should not be ruled out, as there often are lag effects with respect to the influence of climate on disease dynamics in wildlife. Some text on this possibility could be added to the Discussion.

Response: This is an important point. However, our findings do not suggest that there is no correlation between microclimatic variation and Bd infection, but rather that there was little variation in microclimatic conditions among plots (as expected, given their close proximity and similar habitat characteristics). Therefore, we do not expect spatial variation in infection to be generally driven by environmental conditions at this spatial scale in this system. To make this point clearer, we have rephrased the sentence at L162-165:

“Thus, although climatic factors frequently influence amphibian–Bd host–parasite dynamics^{20,21}, our results indicate that the fine-scale spatial clustering of Bd infection observed in this system is unlikely to be driven by spatial variation in environmental conditions.”

Further, stochasticity receives little to no attention as a potential driver of spatial variation in infection rates. There are relatively small sample sizes in the study, and the IBM starts in simulation with only one individual becoming infected with Bd. This individual then infects others, which can lead to high infection rates. Given this, the stochasticity involved with respect to one individual becoming infected could play a large role in the spatial variation in infection rates among subpopulations. Thus, more discussion of the role of randomness or chance events in infection dynamics is warranted in the main text.

Response: Thank you for this comment. In our analyses of the empirical data, observation-level stochasticity and sample size were accounted for through the use of statistical models. Regarding the reviewer’s statement that “this individual then infects others, which can lead to high infection rates”, we agree and consider this to be the main driver of infection dynamics at the subpopulation scale (rather than among-subpopulation differences in environmental conditions or host susceptibility), as supported by the empirical data and IBM results. In the IBM we intentionally initiated simulations with a single infected individual, to demonstrate that even one infection at t_0 can typically produce substantial population depression under the mean infection parameter estimates from RFC and HUI. In exploratory analyses using higher initial prevalences, as expected, population depression increased while the likelihood of Bd fade-out was reduced. The effects of stochasticity are discussed in the legend of Figure 4, where we wrote:

“The non-linear relationship observed in the contour lines in (a) arise because at very low λ_{rd} , stochastic infection events cause large relative impacts on δ , leading to high population depression when compared to a no-Bd baseline. As λ_{rd} increases, the effect of stochasticity is reduced, resulting in a temporary plateau in depression. However, once the host density is high for the pathogen to invade, sustained Bd transmission drives further population depression, and as λ_{rd} increases and Bd transmission approaches saturation a lower value of $\lambda_{syntopic}$ is required to produce an equivalent population depression.”

These results show, as expected from statistical theory, that stochasticity in infection dynamics plays a more important role when host population density is low.

Considering this, we believe that adding further discussion of stochasticity lies beyond the scope of our study and may add confusion for readers rather than improving the clarity of our findings.

The authors often focus reporting of infection dynamics as a percentage of detected infections (e.g., Line 120). Given the role of spatial proximity in infection dynamics, it would be beneficial for the reader to have a better understanding of both overall subpopulation abundance and pathogen prevalence (number of Bd positive / total number tested), rather than data on the infections only. Indeed, reporting on the abundance estimates and pathogen prevalence results within the main text would result in a more complete manuscript for the reader.

Response: Thanks for this helpful suggestion. The range of mean abundance values has been provided in the text in L375-377. The number of total animals tested / total animals Bd positive is provided in L430-435:

“At RFC, the number of *R. darwinii* captures and Bd-positive samples per plot were: RFC1 (130/19), RFC2 (99/9), RFC3 (129/5), RFC4 (271/1), RFC5 (240/39), and RFC6 (85/3), totalling 76 positives. At HUI, the corresponding values were: HUI1 (57/0), HUI2 (114/1), HUI3 (142/10), HUI4 (181/2), HUI5 (154/0), and HUI6 (118/0), totalling 13 positives. At Parque Tantauco, the corresponding values were TAN1 (592/9) and TAN2 (360/4), totalling 13 positives.”

Minor Comments

When referring to disease outbreaks in wildlife, epizootic often is the preferred term in the ecological literature, rather than epidemic. Consider changing epidemic to epizootic throughout the manuscript.

Response: Thank you for this comment. While we agree that epizootic is often used in wildlife disease research, including studies on Bd, we believe that the broader term epidemic is more widely used in disease ecology, which is the main focus of this paper. The Greek root “demos” is commonly translated as “people,” but also as “population”, and terms such as epidemiology and demography are broadly applied to populations of any species, human or non-human. For consistency, we prefer to use a single term that can encompass all populations, since the processes described by epidemic and epizootic are the same. Supporting this usage, a search in *Nature Ecology & Evolution* returned 51 results for “epidemic” but only 4 for “epizootic.” We have therefore retained the use of epidemic throughout the manuscript.

I wonder how the methodology of this study could be applied to learn about fine-scale patterns of Bd infection dynamics in aquatic/semi-aquatic amphibians? This would make for an impactful Discussion paragraph and inform future studies aiming to conduct this type of work in various systems.

Response: This is a very interesting point. In our view, the main difficulty is that in most studies (especially those on pond-breeding species) monitoring is typically restricted to breeding sites. Often, the study area is not well defined, and detailed tracking of animal space use is lacking, at least when using capture–recapture methods. Therefore, we believe our methodology (search-encounter capture–recapture data) is most applicable to fully terrestrial species, but difficult to implement for pond-breeding amphibians. A similar approach to ours could be applied to stream-breeding amphibians, and some researchers already collect spatial data when monitoring these species, though such data are rarely incorporated into capture–recapture inference. We see an opportunity for these studies to better integrate spatial information that is already being collected. Although this is a very interesting discussion, we believe it lies beyond the scope of the present study, and adding it here could divert attention from our main message. We are currently preparing a review of demographic studies evaluating the impact of Bd in wild amphibians, where we will provide a detailed discussion of this issue for interested readers.

Thank you for the opportunity to review this manuscript. I hope my comments are helpful to the authors in improving the clarity of their manuscript for readers.

Reviewer #3 (Remarks on code availability):

I found the README file thorough and helpful. As time permitted, I ran through some of the code (some of the main analyses) without any trouble. The scripts were well commented. I was not able to run through every script.

Response: Thank you for your positive feedback on the README file and code. We are glad to hear that the main analyses ran smoothly and that the scripts were clear. We appreciate the time you dedicated to testing the code.

Response to Reviewer Comments

Reviewer #1

Remarks to the Author:

I believe the authors have done an excellent job of revising their manuscript and providing detailed point-by-point responses to reviewer comments. My comments are very minor and are only suggestions for the author's consideration.

Response: We sincerely thank the reviewer for their positive and constructive feedback throughout the review process. We greatly appreciate their thoughtful suggestions and supportive assessment of our revised manuscript.

A few minor points for consideration:

L32: I know Bd is generally considered an 'aquatic pathogen', as it is referred to here. But I think your results indicate that it is not exclusively an aquatic pathogen and perhaps we need to stop referring to it as such.

Response: We understand the reviewer's point of view; however, we consider that our study does not demonstrate that Bd is not an aquatic pathogen. In our study system, infection prevalence is relatively low under most circumstances—similar to what has been observed in other fully terrestrial amphibians—suggesting a lower contact rate with the infective stage of Bd (an aquatic zoospore) compared with species that use water bodies during part of their life cycle (as discussed in L333–334). Conversely, we argue that our study provides evidence that, in fully terrestrial amphibians, Bd behaves more like a directly transmitted pathogen, where contact rates between infected and susceptible individuals—often correlated with host density—are critical for transmission (discussed in L318–323). Therefore, we believe that referring to Bd as an aquatic pathogen remains consistent with the available evidence.

L84: I really like this change to include this information on terrestriality in amphibian species up front. It helps to better highlight the novelty of the empirical contribution provided by this paper.

Response: We are glad the reviewer found this addition helpful.

L226 (paragraph): This is very interesting. In the systems I work in in Australia, an analogous situation seems to be occurring.

Response: This is very interesting, thank you for sharing.

L251: Suggest: "This indicates Bd has the capability to drive rapid collapse in wild *R. darwinii* populations, consistent with our observations from TAN1 and TAN2."

Response: Thank you for this helpful suggestion, we have made the change as recommended.

L296: The phrasing is a bit strange here. I suggest: “Our findings reinforce the universal variability of patterns and their underlying mechanisms across spatial scales in natural systems”.

Response: Thank you for this helpful suggestion, we have rephrased this as follows:

“Our findings are consistent with and further illustrate the universal variability of patterns and their underlying mechanisms across spatial scales in natural systems”.

L338: Thanks for this addition. Maybe an added detail that could be useful in this new sentence is that slower declines in terrestrial species are perhaps more likely at low densities. Your results show that Bd can clearly drive rapid declines in some circumstances in a terrestrial species. (adding this detail will also help better situate this sentence in the context of the rest of the information presented in this paragraph).

Response: We thank the reviewer for this thoughtful suggestion. We have updated this sentence to:

“Due to potentially reduced transmission rates, the rate of decline may be slower in fully terrestrial species than in aquatic ones, particularly when the abundance of the focal terrestrial host or syntopic amphibians is moderate to low.”

Reviewer #2

Remarks to the Author:

The authors provided a revised version of their manuscript assessing spatial epidemiology (chytridiomycosis) on the critically endangered Darwin's frog. In this new version the authors have addressed successfully most of my concerns. I find the study suitable for publication. Best regards!

Remarks on code availability:

I reviewed the code during my first review and it worked well.

Response: We sincerely thank the reviewer for their positive feedback and for the constructive comments provided during the first review, which greatly improved our manuscript. We truly appreciate their time, expertise, and supportive assessment of our revised version.

Reviewer #3

Remarks to the Author:

I very much appreciate the authors for considering each of my comments and revising their manuscript accordingly. When there was disagreement, I found the authors rebuttals thoughtful and respectful, and I do not feel it necessary to reiterate or push back on any previous suggestions that were not incorporated. Further, I do not have any remaining concerns about the manuscript nor the analyses. I believe the authors have addressed each of my main concerns. As presented, I believe this work on variation in infection dynamics within populations (i.e., fine-scale variation in infection rates) will advance how we think about pathogen transmission moving forward, and the methodology used will be applied broadly in disease ecology.

More specific to the revisions, I felt the authors did a nice job better incorporating key information from the supplement into the main text, which provided necessary context and resulted in a more comprehensive manuscript. I also think the authors did a nice job reframing and focusing the beginning of the Introduction. Overall, it appears that the authors did a thorough job in addressing other reviewer comments in addition to considering each of my recommendations. Overall, I congratulate the authors on a nice study. It was a pleasure to review.

Remarks on code availability:

The README file is helpful and I was able to run much of the code. The methodology is robust and has high potential to be useful to others aiming to adopt some of these analytical approaches in their own work.

Response: We sincerely thank the reviewer for their thoughtful and encouraging feedback, both in the first and second rounds. We greatly appreciate the time and care taken to provide such constructive comments, which have helped strengthen the manuscript. We are especially grateful for the reviewer's positive assessment of our revisions and their kind remarks about the study's contribution and methodological approach.